



Hydrology and
Earth System
Sciences

# Revisiting extreme precipitation amounts over southern South America and implications for the Patagonian Icefields

**Tobias Sauter**

Climate System Research Group, Institute of Geography, Friedrich-Alexander-Universität Erlangen-Nürnberg (FAU), Erlangen, Germany

**Correspondence:** Tobias Sauter (tobias.sauter@fau.de)

**Abstract.** Patagonia is thought to be one of the wettest regions on Earth, although available regional precipitation estimates vary considerably. This uncertainty complicates understanding and quantifying the observed environmental changes, such as glacier recession, biodiversity decline in fjord ecosystems and enhanced net primary production. The Patagonian Icefields, for example, are one of the largest contributors to sea-level rise outside the polar regions, and robust hydroclimatic projections are needed to understand and quantify current and future mass changes. The reported projections of precipitation from numerical modelling studies tend to overestimate those from in situ determinations, and the plausibility of these numbers has never been carefully scrutinized, despite the significance of this topic to our understanding of observed environmental changes. Here I use simple physical arguments and a linear model to test the plausibility of the current precipitation estimates and its impact on the Patagonian Icefields. The results show that environmental conditions required to sustain a mean precipitation amount exceeding $6.09 \pm 0.64 \, \mathrm{m \, yr^{-1}}$ are untenable according to the regional moisture flux. The revised precipitation values imply a significant reduction in the surface mass balance of the Patagonian Icefields compared to previously reported values. This yields a new perspective on the response of Patagonia's glaciers to climate change and their sea-level contribution and might also help reduce uncertainties in the change of other precipitation-driven environmental phenomena.

## 1 Introduction

Patagonia's weather and climate are largely shaped by baroclinic eddies, which are characterized by the interaction of the planetary waves with the mean flow (Garreaud, 2009; Garreaud et al., 2013; Schneider et al., 2003; Vallis et al., 2014). The same mesoscale eddies efficiently transfer water vapour from the tropics poleward (Langhamer et al., 2018; Schneider et al., 2010; Trenberth et al., 2005), and regularly (every 9–12 d) these trigger narrow filaments of water-vapour-rich bursts called atmospheric rivers. These features temporarily increase the vertically integrated water vapour content (IWV) in the Southern Hemisphere midlatitudes by more than 200 % (Durre et al., 2006; Waliser and Guan, 2017). More than half of all extreme precipitation events (above the 98th percentile) in Patagonia are associated with landfalling atmospheric rivers (Waliser and Guan, 2017). Given the tight coupling between atmospheric moisture transport and hydroclimatic response, changes in moisture transport mechanisms not only dominate the interannual and multi-decadal precipitation variability in Patagonia (Aguirre et al., 2018; Aravena and Luckman, 2009; Garreaud, 2007; Garreaud and Muñoz, 2005; Muñoz and Garreaud, 2005; Sauter et al., 2009; Schneider and Gies, 2004; Viale and Garreaud, 2015; Weidemann et al., 2013, 2018a) but also dictate the fate of the ice masses in this region.

The Andes constitute an effective barrier to the impinging moist tropospheric air masses, forming one of the most extreme climatic divides found worldwide (Barrett et al., 2009; Garreaud, 2009; Garreaud et al., 2013; Rasmussen et al., 2007; Smith and Evans, 2007). The strong orographic influence on the precipitation distribution is evident from both remote sensing (Wentz et al., 1998) and terrestrial observa-

**Table 1.** Comparison of mean precipitation estimates on the SPI and NPI averaged over the entire icefield and the western (210–330°) and eastern (30–150°) slopes. Values are given in m w.e. $yr^{-1}$. The local maximum values, if available, are shown in parentheses. OPM: linear orographic precipitation model. DR: drying ratio.

| | SPI | | | NPI | | | Periods |
|---|---|---|---|---|---|---|---|
| | Mean | West | East | Mean | West | East | |
| $OPM_{0.45}$ | $5.06 \pm 0.51$ | $5.93 \pm 0.60$ | $4.06 \pm 0.42$ | $5.38 \pm 0.59$ | $5.83 \pm 0.64$ | $4.37 \pm 0.48$ | 2010–2016 |
| | $(10.09 \pm 0.92)$ | $(10.09 \pm 0.92)$ | $(9.92 \pm 0.95)$ | $(9.43 \pm 0.93)$ | $(9.43 \pm 0.93)$ | $(9.30 \pm 0.92)$ | |
| > 3000 m | $8.03 \pm 0.81$ | $8.60 \pm 0.85$ | $8.10 \pm 0.82$ | $7.16 \pm 0.79$ | $7.40 \pm 0.78$ | $6.66 \pm 0.73$ | |
| 2500–3000 m | $6.37 \pm 0.65$ | $6.93 \pm 0.70$ | $6.13 \pm 0.63$ | $6.58 \pm 0.67$ | $6.84 \pm 0.70$ | $5.58 \pm 0.53$ | |
| 2000–2500 m | $5.39 \pm 0.54$ | $5.70 \pm 0.58$ | $4.93 \pm 0.50$ | $5.69 \pm 0.58$ | $6.20 \pm 0.63$ | $5.10 \pm 0.52$ | |
| 1000–2000 m | $5.29 \pm 0.54$ | $6.13 \pm 0.62$ | $4.26 \pm 0.44$ | $5.58 \pm 0.62$ | $5.77 \pm 0.64$ | $4.81 \pm 0.53$ | |
| < 1000 m | $4.26 \pm 0.44$ | $5.43 \pm 0.56$ | $3.04 \pm 0.32$ | $4.81 \pm 0.54$ | $5.77 \pm 0.64$ | $3.05 \pm 0.35$ | |
| $OPM_{0.60}$ | $5.99 \pm 0.59$ | $7.02 \pm 0.68$ | $4.80 \pm 0.49$ | $6.09 \pm 0.64$ | $6.60 \pm 0.69$ | $4.90 \pm 0.53$ | 2010–2016 |
| | $(11.58 \pm 0.98)$ | $(11.58 \pm 0.98)$ | $(11.39 \pm 0.99)$ | $(10.37 \pm 0.96)$ | $(10.37 \pm 0.96)$ | $(10.12 \pm 0.95)$ | |
| > 3000 m | $8.89 \pm 0.89$ | $9.56 \pm 0.94$ | $8.94 \pm 0.90$ | $7.67 \pm 0.85$ | $7.93 \pm 0.85$ | $7.07 \pm 0.79$ | |
| 2500–3000 m | $7.09 \pm 0.73$ | $7.73 \pm 0.78$ | $6.81 \pm 0.71$ | $7.05 \pm 0.73$ | $7.35 \pm 0.75$ | $5.92 \pm 0.59$ | |
| 2000–2500 m | $6.08 \pm 0.61$ | $6.46 \pm 0.65$ | $5.55 \pm 0.57$ | $6.16 \pm 0.64$ | $6.75 \pm 0.69$ | $5.48 \pm 0.57$ | |
| 1000–2000 m | $6.19 \pm 0.61$ | $7.17 \pm 0.70$ | $5.00 \pm 0.52$ | $6.21 \pm 0.67$ | $6.45 \pm 0.70$ | $5.30 \pm 0.58$ | |
| < 1000 m | $5.34 \pm 0.53$ | $6.77 \pm 0.66$ | $3.84 \pm 0.39$ | $5.74 \pm 0.58$ | $6.84 \pm 0.68$ | $3.72 \pm 0.40$ | |
| $DR_{0.45}$ | 4.67 (8.06) | 4.66 (7.98) | 4.70 (7.95) | 4.94 (9.68) | 4.95 (9.68) | 5.08 (9.27) | 2010–2016 |
| $DR_{0.60}$ | 8.45 (17.71) | 8.41 (17.49) | 8.53 (17.40) | 9.16 (22.12) | 9.19 (22.12) | 9.54 (20.99) | |
| Other studies | | | | | | | |
| Schaefer et al. (2015) | 8.36 | | | $8.03 \pm 0.37$ | | | 1975–2011 |
| | (> 20.0) | | | (> 15.0) | | | |
| Mernild et al. (2017) | $8.13 \pm 0.32$ | | | $6.95 \pm 0.34$ | | | 1979–2014 |
| | (> 15.0) | | | (> 15.0) | | | |
| Lenaerts et al. (2014) | – | | | – | | | 1979–2012 |
| | (> 30.0) | | | (> 30.0) | | | |
| Escobar (1992) | 7.0 | | | 6.7 (over the broad plateau) | | | 1960–1980 |

tions (Fig. 1). Despite observational uncertainty along the coast, two characteristic precipitation regions are apparent: (i) a maritime Precordillera region with annual precipitation exceeding 2–3 m w.e. (water equivalent) and (ii) a semi-arid rain-shadow region (< 0.5 m w.e.) east of the main ridge that extends several thousand kilometres towards the South Atlantic. However, little is known about precipitation along the main ridge and, in particular, on the Patagonian Icefields. Current estimates from firn cores (Schwikowski et al., 2006; Shiraiwa et al., 2002), discharge measurements (Escobar, 1992) and numerical modelling (Bravo et al., 2019; Lenaerts et al., 2014; Mernild et al., 2017; Schaefer et al., 2013, 2015; Weidemann et al., 2018b) suggest average annual precipitation rates of 5 to 8 m w.e. $yr^{-1}$ and of 7 to > 10 m w.e. $yr^{-1}$ for the Northern and Southern Patagonian Icefield (NPI; SPI), respectively (see Table 1). Extreme precipitation rates between 15 m w.e. $yr^{-1}$ (Mernild et al., 2017; Schaefer et al., 2013, 2015; Schwikowski et al., 2006) and 30 m w.e. $yr^{-1}$ are suspected at isolated locations (Lenaerts et al., 2014). If these precipitation magnitudes are realistic, it is likely that the SPI is one of the wettest – if not *the* wettest – places on Earth.

The considerable uncertainty in precipitation amounts in Patagonia not only affects our current understanding of the local hydrological cycle but also has profound impacts on studies concerned with fjord ecosystems (Landaeta et al., 2012), biological production in water columns (Aracena et al., 2011; Vargas et al., 2018), net primary production (Jobbágy et al., 2002), glacier mass balance (Escobar, 1992; Foresta et al., 2018; Lenaerts et al., 2014; Mernild et al., 2017; Schaefer et al., 2013, 2015; Schwikowski et al., 2006; Shiraiwa et al., 2002; Weidemann et al., 2018b; Willis et al., 2012) and its contribution to sea-level rise (Braun et al., 2019; Malz et al., 2018; Marzeion et al., 2012; Rignot et al., 2003). Reducing the plausible range of precipitation rates is a key step towards improved process understanding of such systems and offers new perspectives on future changes.

Here I use simple physical scaling arguments and a linear modelling approach to test the plausibility of the current precipitation estimates in central Patagonia (45–52° S). In particular, I address the question of whether the water vapour flux (WVF) from the tropics to the midlatitudes by baroclinic eddies can sustain these extreme precipitation estimates. The assessment of the hypothesis relies on three fundamental as-

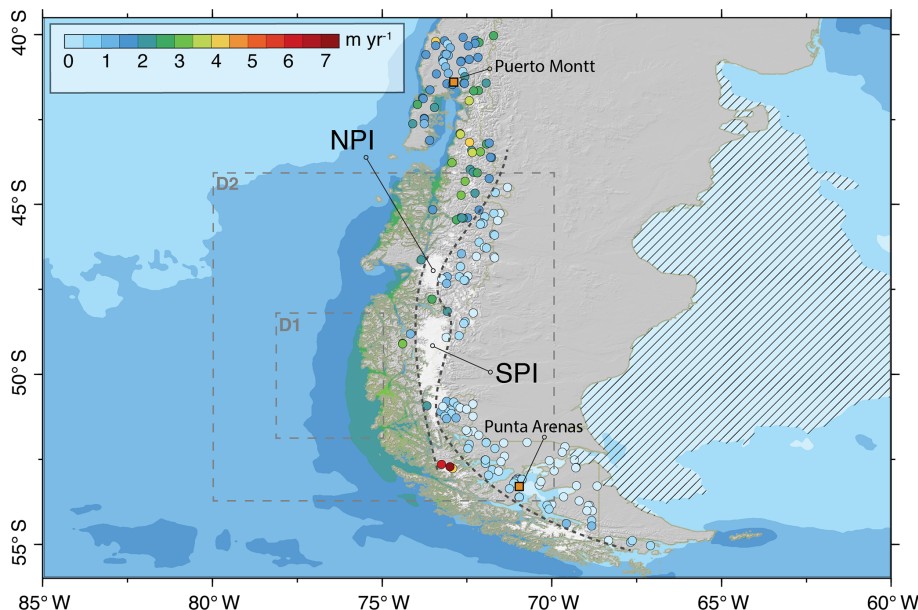

**Figure 1.** Precipitation climatology in southern South America. The filled-in circles indicate precipitation amounts observed by the observational network of the Dirección Meteorológica de Chile (DMC) and the Dirección General de Aguas (DGA). Also included are the permanent weather station measurements that were taken at the Gran Campo Nevado Ice Cap. The colour-shaded areas over the ocean show the rainfall distribution based on the Global Precipitation Measurement (GPM) satellite mission. Black dashed lines roughly delineate the maritime Precordillera range, Andes main ridge and the semiarid Pampa region. Also indicated are the Northern (NPI) and Southern Patagonian Icefields (SPI). The dashed area shows the semi-arid rain-shadow region. Also shown are the simulation (D2) and forcing (D1) domains.

sumptions. (i) The orographically induced precipitation is proportional to the incoming WVF, which acts as the major moisture resource for the precipitation system. This implies that uncertainties in the incoming WVF directly impact the precipitation estimate. (ii) The terrain-forced uplift and condensation of moist air masses is assumed to be the dominant precipitation formation process in central Patagonia. (iii) The atmospheric drying ratio (DR) derived from observed isotope data is a valid measure for the cross-mountain fractionation of the WVF. Based on this assumption, the proposed methods are constrained by the DR to accurately reproduce the fraction of the water vapour flux removed by orographic precipitation.

After a description of the methods (Sect. 2), the moisture transport and its role on local precipitation formation in southern South America is explored in more detail (Sect. 3.1). The next chapter (Sect. 4) begins with the assessment of the precipitation estimates (Sect. 4.1) and discusses its implications for the surface mass balance of the Patagonian Icefields (Sect. 4.2). We will further link the surface mass balance to the local hydrological cycle to understand the long-term evolution of glaciers in this region (Sect. 4.3). Following this section, the limitation and uncertainty of the proposed approach is discussed (Sect. 4.4). The last section provides a conclusion of the main findings.

## 2 Methodology

### 2.1 DR scaling (DRS)

To provide a first assessment of the magnitude of precipitation, mean precipitation is estimated along the western slopes of the Andes (45–52° S and 73–76° W) using a simple DR scaling (DRS). The DR in Patagonia, defined as the fraction of the WVF removed by orographic precipitation, is known to be the highest ($\sim 0.45$–$0.5$) worldwide (Mayr et al., 2018; Smith and Evans, 2007). The ratio is a characteristic measure for mountain ranges and is independent of the incoming WVF. If the WVF and the DR are known, one can estimate the mean homogeneous (uniform) precipitation amount. To add altitude-dependent precipitation variability, the amount was redistributed mass consistently by optimizing the vertical precipitation gradient using a Newton–Raphson algorithm (Press et al., 2007). The lapse-rate optimization finds the roots of the function

$$F(\gamma) = \frac{(D_0 \cdot F_0)}{A} - \iint_D (P_0 + \gamma \cdot h) = 0, \quad (1)$$

where $A$ (m$^2$) is the study domain area, $P_0$ (m) is the background precipitation at sea level, $\gamma$ (m m$^{-1}$) is the precipitation gradient and $h$ (m) is the terrain height. The first term on the right represents the potential precipitation resulting from the WVF, $F_0$ (kg m$^{-1}$ s$^{-1}$), and the given DR, $D_0$ (–). The

second term is the precipitation integrated over the domain $D$ resulting from the linear interpolation. This interpolation via lapse rate converts the entire specified WVF fraction into precipitation regardless of the saturation vapour deficit of the impinging air masses. However, orographic precipitation can only occur when the terrain-forced uplift and cooling of air masses lead to water vapour condensation. To take this condition into account, only lower-tropospheric (below 950 hPa) air masses are considered with a relative humidity equal to or exceeding 90 % (Jarosch et al., 2012; Weidemann et al., 2013). The DRS provides a first-order approximation but neglects heterogeneity and important processes such as airflow dynamics and cloud physics.

## 2.2 Linear orographic precipitation model (OPM)

To account for these aspects, a set of realistic and extreme ensemble experiments has been designed using a linear orographic precipitation model (OPM), which represents many processes, such as condensation and hydrometeor conversion, using relatively simple formulations for airflow dynamics and cloud physics (e.g. Garreaud et al., 2016; Jarosch et al., 2012; Smith and Barstad, 2004; Smith and Evans, 2007; Weidemann et al., 2018a). The model builds upon the original formulation of the linear orographic precipitation model (Barstad and Smith, 2005; Smith and Barstad, 2004), including a correction of the WVF downstream (Smith and Evans, 2007) and an optimization to enforce the model towards a given drying ratio. It solves two steady-state advection equations describing the change in the vertically integrated cloud water density and hydrometeors density due to advection, condensation of water vapour by terrain-forced uplift, conversion from cloud water to hydrometeors and hydrometeor fallout. Mountain wave theory allows for the decay of the vertical velocity caused by tilting mountain waves and consequently constrains the water vapour condensation rate. Assuming horizontal uniform background flow and properties (e.g. atmospheric stability), the orographic precipitation can be represented by a transfer function of

$$\hat{P}(k,l) = \frac{C_\mathrm{w} i \sigma \hat{h}(k,l)}{(1 - imH_\mathrm{w})(1 + i\sigma\tau_\mathrm{c})(1 + i\sigma\tau_\mathrm{f})}, \quad (2)$$

where $C_\mathrm{w}$ (–) is the uplift sensitivity factor, which relates the vertical air motion to the condensation rate; $i$ is the imaginary unit; $\sigma = Uk + Vl$ is the intrinsic frequency, where $k$ and $l$ are the horizontal wavenumbers; $\hat{h}(k,l)$ is the Fourier transform of the terrain; $m$ is the vertical wavenumber; $H_\mathrm{w}$ (m) is the water vapour scale height; and $\tau_\mathrm{c}$ and $\tau_\mathrm{f}$ (s) are the timescales for the conversion from cloud water to hydrometeors and their precipitation. The airflow dynamics is represented by the vertical wavenumber $m$, which is a function of the atmospheric stratification represented by the moist Brunt–Väisälä frequency $N_\mathrm{m}^2$ (s$^{-1}$). Thus, the parsimonious model contains five parameters: the uplift sensitivity factor $C_\mathrm{w}$, the moist buoyancy frequency $N_\mathrm{m}^2$, the water vapour

scale height $H_\mathrm{w}$, and the condensation and fallout timescales $\tau_\mathrm{c}$ and $\tau_\mathrm{f}$. The mean horizontal wind velocities ($U$, $V$) and the parameters $C_\mathrm{w}$ and $N_\mathrm{m}^2$ are calculated from 6-hourly ERA-Interim (ECMWF Reanalysis) fields (2010–2016) below the 500 hPa geopotential level off Patagonia's western coast between 48 and 52° S and 75 and 78° W (Fig. 1, D1; Smith and Barstad, 2004). Contrary to most other studies, $H_\mathrm{w}$ is directly derived from the incoming WVF, $F_0$ (kg m$^{-1}$ s$^{-1}$), using $H_\mathrm{w} = F_0/(\rho q_\mathrm{w} U)$, where $q_\mathrm{w}$ (kg kg$^{-1}$) is the total mixing ratio and $\rho$ (kg m$^{-3}$) the air density. The timescales of $\tau_\mathrm{c} = \tau_\mathrm{f} = 850$ s are fixed for all experiments, which are realistic values for the southern Andes and produce remarkable similar results to numerical models (Garreaud et al., 2016; Smith and Evans, 2007). The total precipitation field in physical space,

$$P(x,y) = \max\left[\left(\iint \hat{P}(k,l)e^{i(kx+ly)}\mathrm{d}k\mathrm{d}l + P_\infty\right), 0\right], \quad (3)$$

is finally obtained by double Fourier transform Eq. (2) and adding the synoptic-scale background precipitation $P_\infty$ (m) followed by the truncation of negative values. For consistency, $P_\infty$ is calculated by removing the orographic component from the ERA-Interim precipitation field (for details see Dee et al., 2011, and Jarosch et al., 2012). To enforce the model towards a given drying ratio $D_0$, $P_\infty$ is scaled by a constant so that the calculated DR corresponds to $D_0$. The model is solved on a 90 m Shuttle Radar Topography Mission (SRTM) dataset, resampled at 1 km resolution (Jarvis et al., 2008).

## 2.3 Experiments

Within the scope of this study, two experiments each were performed with the DRS and the OPM. In the first experiment it was tested whether a combination of "realistic" atmospheric environmental conditions (derived from the reanalysis data), observed DR of 0.45 (Mayr et al., 2018) and WVF, provide the basis to sustain the precipitation estimates of previous studies. The second experiment delivers an "extreme" scenario by setting the DR to a higher value of 0.6. In this OPM experiment, the buoyancy sensitivity factor ($C_\mathrm{w} = 0.004$) and the moisture stability frequency ($N_\mathrm{m} = 0.007$ s$^{-1}$) were also set to their 98th percentile values. The extreme scenario thus represents atmospheric conditions that exist in nature, but whose occurrence is extremely rare. To obtain an upper limit of the precipitation potential, we assume that this atmospheric condition is present every day. Ensemble experiments were created with the OPM for both scenarios. Each ensemble comprises 40 ensemble members. The ensemble members consider the uncertainty of the initial state in wind direction and moisture content by randomly perturbing $U$ (5 %), $V$ (5 %) and $H_\mathrm{w}$ (10 %) around their mean value. From here on the realistic simulations are marked with the subscript 0.45 (DRS$_{0.45}$ and

OPM$_{0.45}$), while the extreme simulations are marked with the subscript 0.60 (DRS$_{0.60}$ and OPM$_{0.60}$).

## 2.4 Atmospheric simulations using the Weather Research and Forecast (WRF) model

To analyse the influence of nonlinear-flow regimes on precipitation patterns, atmospheric simulations were performed with the Weather Research and Forecast (WRF) model, version 3.8.1. The model was configured with three one-way nested domains with a horizontal resolution of 12.5 km, 2.5 km and 500 m, which were centred over the Southern Patagonian Icefields. The model configuration and parameterizations used in this study are shown in Table S4 in the Supplement. To achieve the required resolution in the inner domain, the standard terrain data were replaced by NASA Shuttle Radar Topography Mission (SRTM) data (https://cgiarcsi.community/data/srtm-90m-digital-elevation-database-v4-1, last access: 16 November 2016). Furthermore, the land use classification was updated with the ESA CCI (Climate Change Initiative) dataset (https://www.esa-landcover-cci.org, last access: 8 February 2018). This way the glacier outlines could be improved significantly. The outermost domain was driven at its lateral boundaries by the ERA-Interim reanalysis dataset with a spatial resolution of $0.75° \times 0.75°$ in longitude and latitude and a time interval of 6 h. With the above setup, individual events were calculated with WRF. Each simulation had a spin-up of at least 12 h.

## 3 Results

### 3.1 Moisture transport

Observations of IWV and WVF are sparse in South America and limits the analysis of the moisture transport to a few locations (see Fig. 2). The only available soundings for the region are Puerto Montt (41.4347° S, 73.0975° W) on the Pacific coast and Punta Arenas (53.0033° S, 70.8450° W) located at the Strait of Magellan (Durre et al., 2006). Along the coast at the latitude of Puerto Montt, the average WVF in the period 1990–2017 was about $165.52 \pm 48.51$ kg m$^{-1}$ s$^{-1}$. Landfalling atmospheric rivers temporarily amplify the WVF by more than 400 kg m$^{-1}$ s$^{-1}$. There is also clear evidence that enhanced atmospheric circulation during strong El Niño events (Ocean Niño Index $> 1.5$) increase the moisture flux over several months (see Fig. 2; e.g. 1997/98). The El Niño signal is less pronounced in Punta Arenas. The atmospheric soundings show opposite linear long-term WVF trends over the period 1990–2016 with a significant ($p < 0.08$) decrease of $-4.46$ kg m$^{-1}$ s$^{-1}$ ($-2.70$ %) per decade in Puerto Montt and a significant ($p < 0.05$) positive trend of 8.79 kg m$^{-1}$ s$^{-1}$ (5.11 %) per decade in Punta Arenas (see Fig. 2). However, change-point analysis shows that the observed WVF trend in Punta Arenas is not constant over time but has shown signifi-

cant abrupt shifts in the past that characterize the transition of water-vapour-rich and water-vapour-poor periods (Killick et al., 2012). A significant transition took place in 2006, which marks the beginning of a relatively water-vapour-rich period (Fig. 2).

The ERA-Interim data, on which the analysis is based, reflect the interannual WVF variability and overall trend of the soundings but slightly overestimates the rate of change in Puerto Montt ($-4.94$ kg m$^{-1}$ s$^{-1}$ per decade; $-4.43$ % per decade) and underestimates the observed trend in Punta Arenas (4.10 kg m$^{-1}$ s$^{-1}$ per decade; 2.70 % per decade). The mean WVF at both sites is weaker than the observed moisture transport. In Puerto Montt, the WVF is about $111.54 \pm 34.40$ kg m$^{-1}$ s$^{-1}$, which is almost 30 % less than the estimate from the atmospheric sounding. The differences between observed WVF ($172.12 \pm 54.19$ kg m$^{-1}$ s$^{-1}$) and re-analysis data ($152.08 \pm 57.08$ kg m$^{-1}$ s$^{-1}$) are much lower in Punta Arenas. It is evident from the soundings that ERA-Interim data are too dry (according to the IWV) in the vicinity of Puerto Montt ($-2.23$ mm, $-14.9$ %, $p < 0.01$) and slightly too wet in the south (0.48 mm, 4.6 %, $p < 0.05$; see Fig. S2 and Table S2). The comparison with atmospheric water vapour data obtained by the Special Sensor Microwave Imager/Sounder (SSMIS) over the ocean confirms the north–south pattern (Wentz et al., 1998; see Fig. S4). While IWV differences between ERA-Interim data and SSMIS south of 45° S are on average smaller than 0.16 mm (1.1 %), larger deficits are apparent north of 45° S ($< -0.8$ mm).

Based on the comparison with the atmospheric soundings and SSMIS observation, ERA-Interim underestimates the IWV along the western coast of Patagonia (D1 in Fig. 1), where the corresponding parameters for the assessment were calculated, by less than 5 %. However, comparison with the soundings suggests that the WVF in the ERA-Interim data along the western coast is weaker by 10 % to 20 % due to uncertainties in moisture advection. In the following analysis, a WVF bias of 10 % is assumed and corrected accordingly.

### 3.2 Physical constraints on local precipitation

To obtain the plausible range of precipitation amounts in central Patagonia, the DRS and the OPM are driven by the ERA-Interim data for the period 2010–2016. The DRS is primarily intended to gain fundamental insights into the order of magnitude of precipitation. As the WVF and the DR (here we use 0.45) are known from ERA-Interim data and isotope observations (Dee et al., 2011; Langhamer et al., 2018; Mayr et al., 2018; Smith and Evans, 2007), one can estimate the mean homogeneous (uniform) precipitation amount using Eq. (1). The mean precipitation at sea level, $P_0$ (m), was taken from the Global Precipitation Measurement (GPM) mission off the shore of the Chilean coast ($\sim 3$ m yr$^{-1}$). Solving the optimization problem (see Eq. 1) resulted in a vertical precipitation gradient of $\sim 0.056$ % m$^{-1}$, which is slightly higher than the previously reported lapse rate of $\sim 0.05$ % m$^{-1}$ (Schaefer

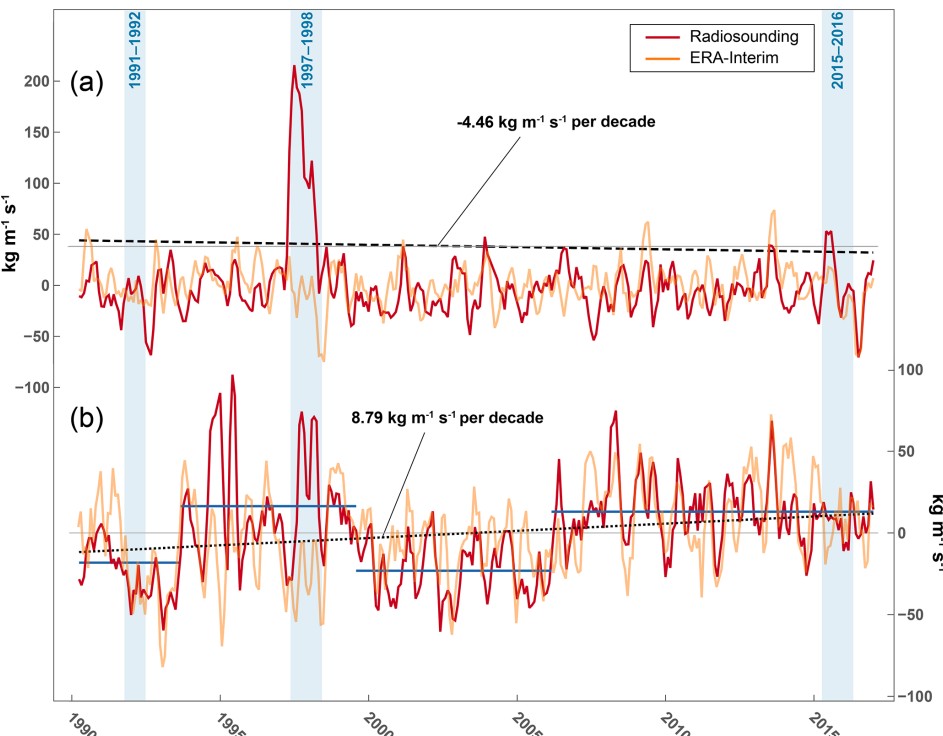

**Figure 2.** Monthly WVF anomalies in Puerto Montt **(a)** and Punta Arenas **(b)**. Shown are the running 3-month mean WVF anomalies for the atmospheric soundings and the nearest ERA-Interim grid point from 1990 to 2016. The blue shaded areas indicate very strong El Niño events (ONI > 1.5). The horizontal blue lines in panel **(b)** show the mean WVF over water-vapour-rich and water-vapour-poor phases.

et al., 2013, 2015). Averaged over the SPI and NPI, this approach produces values of 4.67 and 4.94 m yr$^{-1}$, respectively (see Fig. 1 and Table 1). The highest precipitation amounts are reached at the highest peaks on the NPI with up to 9.68 m yr$^{-1}$. To achieve a DR of 0.6, a precipitation gradient of 0.12 % is required. Such a strong gradient would lead to average precipitation amount of 8.45 and 9.16 m yr$^{-1}$ on the SPI and NPI with maximum values of more than 20 m yr$^{-1}$.

To further include dynamical airflow processes in the estimation, albeit in simplified form, we use the OPM. The OPM is applied to a large domain (Fig. 1, D2) to avoid spurious numerical artefacts. The ensemble mean of the OPM$_{0.45}$ (realistic) experiment gives an average precipitation amount of $5.06 \pm 0.51$ m yr$^{-1}$ over the SPI and $5.38 \pm 0.59$ m yr$^{-1}$ over the NPI (Table 1, Fig. 3), indicating that the WVF can sustain relatively high mean precipitation amounts in Patagonia. However, precipitation estimates are up to 38 % lower than estimates from previous numerical studies (Escobar, 1992; Lenaerts et al., 2014; Mernild et al., 2017; Schaefer et al., 2013, 2015; Schwikowski et al., 2006). The highest mean amounts are found in the highest regions on the western slopes of the icefields (SPI: $5.93 \pm 0.60$ m yr$^{-1}$; NPI: $5.83 \pm 0.64$ m yr$^{-1}$) and at the southernmost end of the SPI. The eastern slopes receive considerably less precipitation (SPI: $4.04 \pm 0.42$ m yr$^{-1}$; NPI: $4.37 \pm 0.48$ m yr$^{-1}$).

The OPM$_{0.60}$ (extreme) experiment shows higher averaged precipitation amounts of $5.99 \pm 0.59$ m yr$^{-1}$ and $6.09 \pm 0.64$ m yr$^{-1}$ at the SPI and NPI, respectively (Fig. 3). The combination of short timescales, a large drying ratio, a strong moist stability frequency and a large uplift sensitivity factor increases the total precipitation and enhances the cross-mountain fractionation. Despite the precipitation-enhancing parameter choices, the maximum precipitation ($11.58 \pm 0.98$ m yr$^{-1}$) represents a reduction of up to 60 % compared to other numerical studies (Lenaerts et al., 2014; Schaefer et al., 2013, 2015).

## 4 Discussion

### 4.1 Assessment of the precipitation estimates

Comparison with in situ observations from the Dirección General de Aguas (DGA; Chile) indicates that the OPM$_{0.45}$ model slightly overestimates precipitation on the eastern (downwind) side by $0.29 \pm 0.37$ m yr$^{-1}$ (see Fig. 5 and Table S3). Larger deviations ($1.07 \pm 1.30$ m yr$^{-1}$) occur at the stations located at the foot of the western slope of the Patagonian Icefields. The overestimation is the result of the rapid increase in model terrain elevation and the absence of nonlinear processes in the OPM (see Sect. 4.4). Please note that this number is somehow misleading, as only three stations are

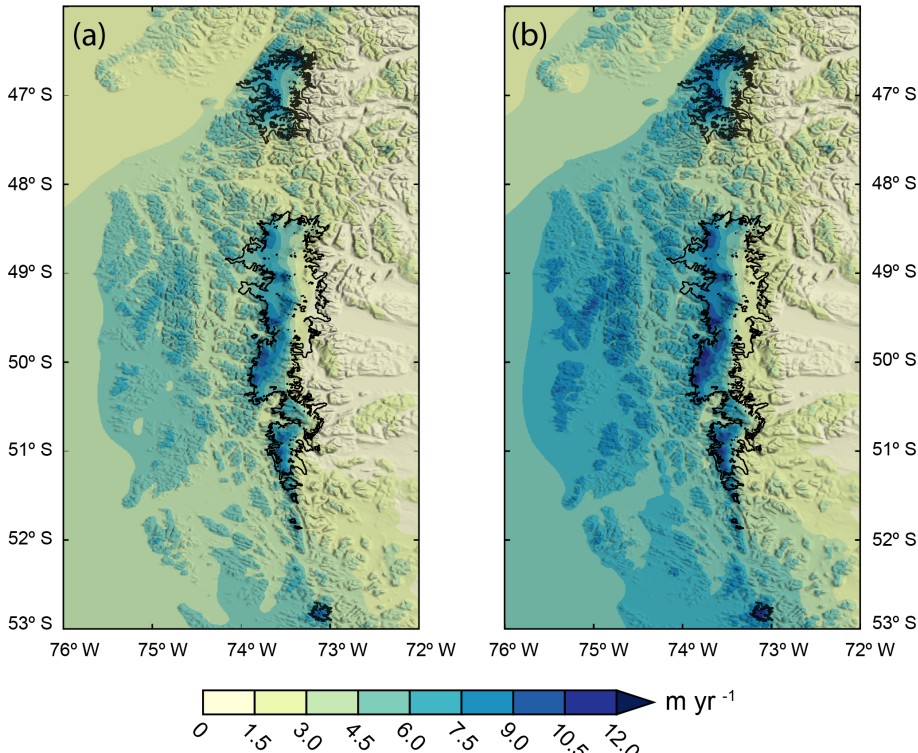

**Figure 3.** Results of the OPM ensemble experiments. Mean precipitation fields (2010–2016) simulated by the OPM using **(a)** the "realistic" ($OPM_{0.45}$) parameter setup and **(b)** the "extreme" ($OPM_{0.60}$) parameter setup using a DR of 0.6 and the 98th percentile values for the uplift sensitivity factor ($C_w = 0.004$) and moist stability frequency ($N_m = 0.007\,s^{-1}$).

available west of the icefields. However, on contrary to the simple DR scaling, the OPM approach captures the observed quick drop in precipitation from west to east (see Fig. 4). Taking all stations into account, the bias between the observations and simulation is about 0.42 m with a root mean squared error (RMSE) of 0.70 m. If the three stations west of the main ridge (Amalia, Puerto Eden and San Rafael glacier) are ignored, the bias is reduced to 0.27 m with an RMSE of 0.44 m. In the $OPM_{0.60}$ experiments, the bias (0.99 m) is significantly higher, indicating that the simulations are much more humid than the observations. The high coefficients of determination suggest that the annual variability is well represented in the $OPM_{0.45}$ as well as in the $OPM_{0.60}$ experiments. In summary, both the temporal variability and the sharp spatial differentiation of precipitation are well represented by the $OPM_{0.45}$ experiment. The $OPM_{0.45}$ result is therefore consistent with the in situ observations, while the $OPM_{0.60}$ result is too wet. It confirms the findings from the isotope measurements that the fractionation of the WVF is in the range of 0.45 (Mayr et al., 2018).

Studies come to very different precipitation totals on the main ridge of the Andes and often diverge even further when it comes to maximum precipitation. The maximum precipitation amount of the $OPM_{0.45}$ experiment ($10.09 \pm 0.92\,m\,yr^{-1}$), found on the SPI plateau, is $\sim 30\,\%$–$70\,\%$

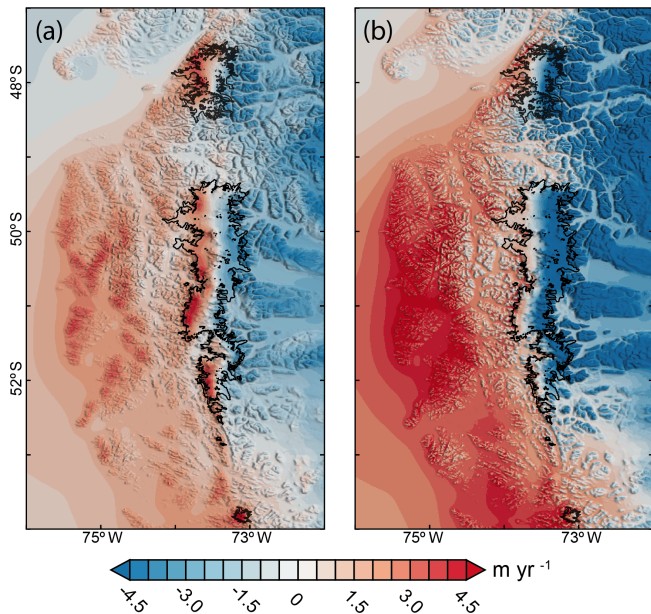

**Figure 4.** Differences between the OPM ensemble experiments and the DR scaling approach. Panel **(a)** shows the differences ($m\,yr^{-1}$) between the $OPM_{0.45}$ and $DRS_{0.45}$ experiment. Similarly, panel **(b)** shows the differences between $OPM_{0.60}$ and $DRS_{0.60}$.

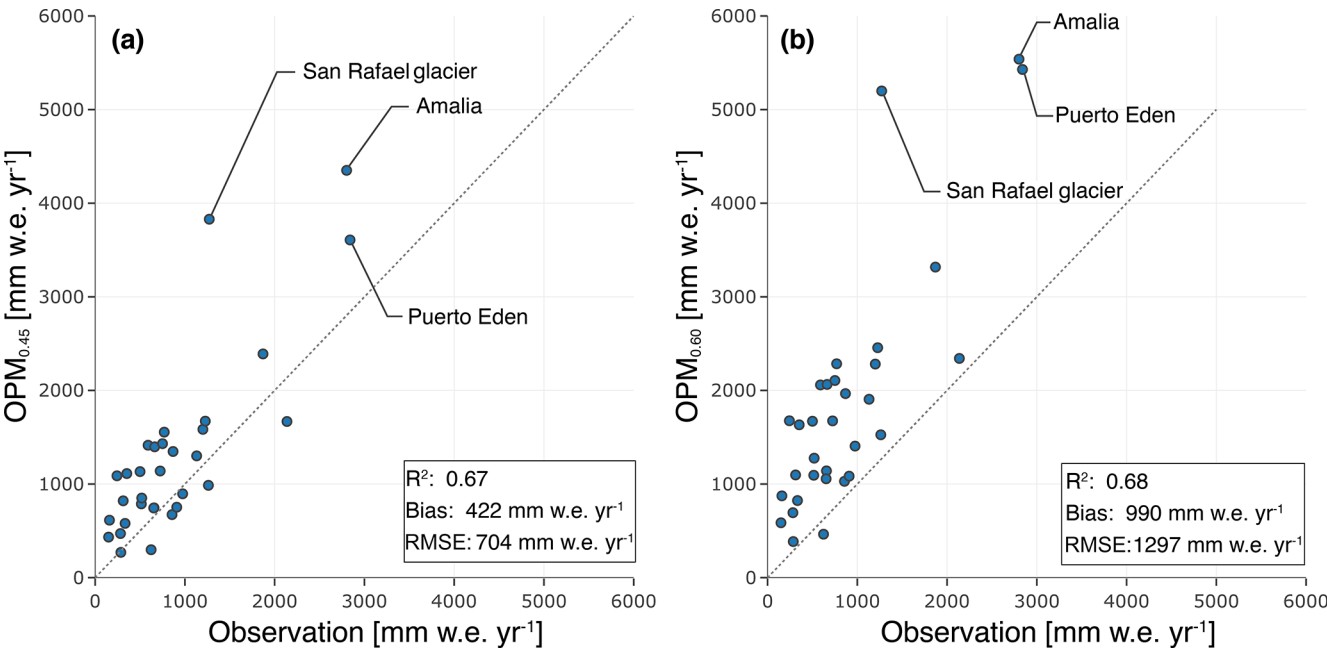

**Figure 5.** Comparison of measured and simulated precipitation for the period 2010–2016. The observations were made by the weather station network of the Dirección Meteorológica de Chile (DMC) and the Dirección General de Aguas (DGA). The only three stations located west of the Patagonian Icefield are labelled.

lower than previously simulated maxima (Lenaerts et al., 2014; Schaefer et al., 2013, 2015) and accumulation rates derived from an ice core (Shiraiwa et al., 2002). The values reported by these studies cannot even be achieved by the OPM$_{0.60}$ experiment, which is still 20 %–60 % lower ($11.58 \pm 0.98$ m yr$^{-1}$). Please note that a snow / rain ratio of 0.55 and a fresh-snow density of 250 kg m$^{-3}$ would still result in fresh-snow accumulation of more than 25 m. The large ensemble spread in maximum values indicates that precipitation is very sensitive to small uncertainties in ambient flow conditions (see Table 1). Even though the uncertainty in the background flow regime and dynamics may also be a possible origin of the extreme precipitation predicted by the mesoscale models, the responsible mechanisms explaining the significant differences remain unclear. It is likely that one reason is the model parameterization of processes. Some microphysical parameterization schemes are more "graupel-friendly" than others, which can lead to strong hydrometeor formation. Since the choice of parameterization combinations can lead to very different results, each model must be examined individually. The sources are manifold and can only be speculative in the context of this study. Given the scarcity of data, especially at higher altitudes, extreme values are difficult to assess. Presumably, the estimated maxima are overestimated due to the extreme parameter choice and to the exclusion of nonlinear effects given the linear nature of the orographic model (see Sect. 4.4).

## 4.2 Consequences of revised precipitation estimates on the surface mass balance of the Patagonian Icefields

These revised precipitation estimates have critical implications for our current understanding of the response of Patagonia's glaciers to climate change. Recent numerical studies (Mernild et al., 2017; Schaefer et al., 2015) suggest a mean annual surface mass gain of $1.78 \pm 0.36$ m to 2.24 m w.e. yr$^{-1}$ for the SPI over recent decades, while surface mass balance (SMB) estimates for the NPI range between $-0.16 \pm 0.73$ and $0.14 \pm 0.49$ m w.e. yr$^{-1}$. However, these assessments used mean precipitation rates well above (40 %–65 %) the plausible range presented in this study.

To quantify the effect of the revised precipitation values on the SMB of the SPI, we use the significant linear relation ($R^2 = 0.96$, $p < 0.05$) between annual snow accumulation and SMB derived from Schaefer et al. (2015) (see Fig. 6), given by SMB $= 1.258 \cdot P_S - 3.935$, where $P_S$ (m) is the mean solid precipitation. The robustness of this relationship is indirectly proven by the study of Mernild et al. (2017), which is very close to the linear fitting line. Taking into account the proposed solid-to-total-precipitation ratio of 0.596 (Schaefer et al., 2015), the mean solid precipitation is $3.02 \pm 0.30$ m w.e. (OPM$_{0.45}$) and $3.57 \pm 0.35$ m. w.e. (OPM$_{0.60}$) for the SPI. Based on this assumption, the revised accumulation values would result in a mean SMB (2010–2016) between $0.56 \pm 0.45$ m w.e. yr$^{-1}$ ($7.82 \pm 6.28$ km$^{-3}$ yr$^{-1}$, OPM$_{0.60}$) and $-0.14 \pm 0.39$ m w.e. yr$^{-1}$ ($-1.95 \pm 5.45$ km$^{-3}$ yr$^{-1}$, OPM$_{0.45}$) on the SPI (Fig. 6). It

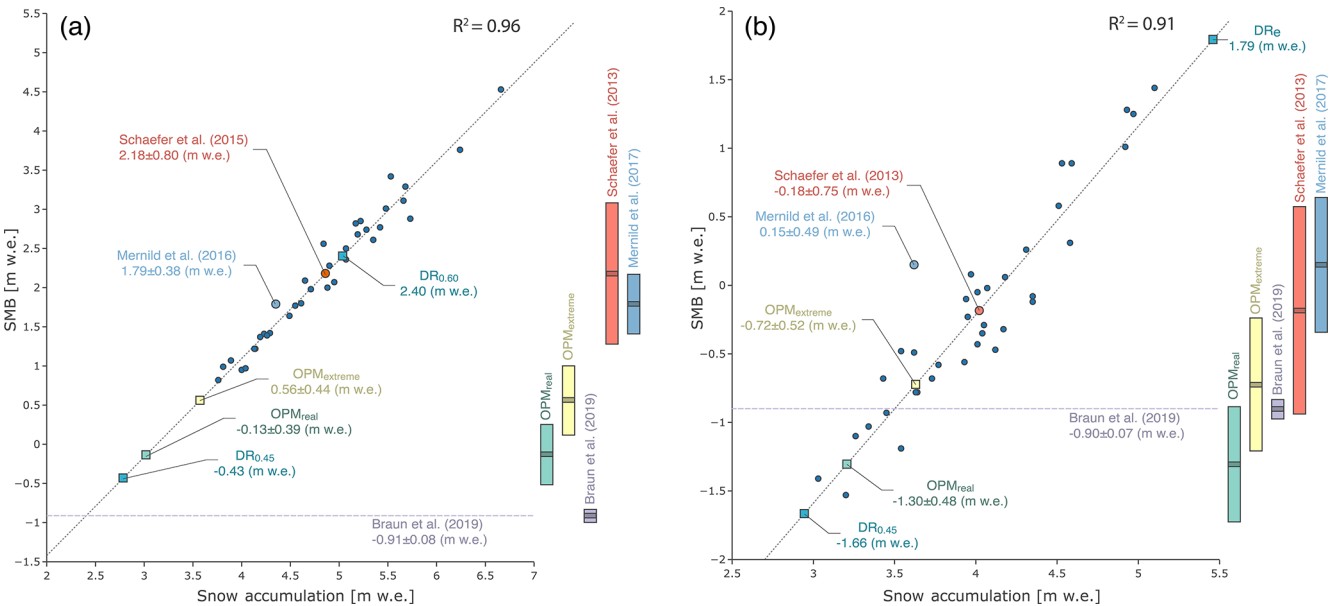

**Figure 6.** Relation between the annual specific accumulation and surface mass balance over the SPI (**a**) and NPI (**b**). The dark blue dots show the annual SMB values from 1975 to 2000 for the SPI and NPI estimated by Schaefer et al. (2013, 2015). The plot also contains the multi-year mean values of Schaefer et al. (2013, 2015), Mernild et al. (2017), and the SMB values derived from this study (labelled dots). The dashed grey horizontal lines show the geodetic mass balances obtained from radar interferometry (Braun et al., 2019). The uncertainty of the individual studies is shown on the right side.

appears that all mean SMB estimates are between the limits of the DRS values (DRS$_{0.45}$: $-0.43$ m w.e. yr$^{-1}$; DRS$_{0.60}$: 1.79 m w.e. yr$^{-1}$). Taking account of the recent geodetic mass balance observations ($-0.941 \pm 0.19$ m w.e.; Malz et al., 2018), the mean mass loss due to calving ranges between $-1.5 \pm 0.64$ m. w.e. yr$^{-1}$ ($-20.95 \pm 8.94$ km$^{-3}$ yr$^{-1}$) and $-0.8 \pm 0.58$ m. w.e. yr$^{-1}$ ($-11.18 \pm 8.10$ km$^{-3}$ yr$^{-1}$). The mean mass balance and calving flows derived here are subject to approach-related uncertainties and may deviate strongly from the values of individual years. A recently published study showed that calving fluxes at Jorge Montt glacier fluctuated between $1.16 \pm 0.66$ and $3.81 \pm 1.10$ km$^{-3}$ yr$^{-1}$ in the years 2012–2018 (Bown et al., 2019). Single extreme events cannot be represented with the approach presented here, since the mean SMB is used together with the geodetic mass balance observations, which also constitutes an integrated value.

The same approach is applied to the NPI to highlight the sensitivity of the SMB to the revised precipitation values. Using the accumulation and SMB data from Schaefer et al. (2013), the linear relationship of SMB $= 1.375 \cdot P_S - 5.713$ between snow accumulation and SMB is obtained. Here we use the same solid-to-total-precipitation ratio, resulting in snow precipitation of $3.20 \pm 0.35$ m w.e. (OPM$_{0.45}$) and $3.61 \pm 0.38$ m. w.e. (OPM$_{0.60}$) for the NPI. When these values are inserted into the linear equation, the mean SMB is $-0.72 \pm 0.52$ m w.e. yr$^{-1}$ ($-3.72 \pm 2.68$ km$^{-3}$ yr$^{-1}$, OPM$_{0.60}$) and $-1.30 \pm 0.48$ m w.e. yr$^{-1}$

($-6.72 \pm 2.48$ km$^{-3}$ yr$^{-1}$, OPM$_{0.45}$). Again, the SMBs derived from the two DRS experiments define the outer limits between which all SMB estimates are located (DRS$_{0.45}$: $-1.66$ m w.e. yr$^{-1}$; DRS$_{0.60}$: 1.79 m w.e. yr$^{-1}$). Comparing the OPM experiments with the geodetic mass balances (Braun et al., 2019) reveals that the SMB of the OPM$_{0.45}$ experiment is lower than the observation ($-0.90 \pm 0.07$ m w.e. yr$^{-1}$). This is an unphysical result which might have two main reasons: (i) the revised precipitation estimates are too low, or (ii) the linear relationship between SMB and snow precipitation is unreliable. The first reason is difficult to verify, but the comparison of the experiments with the stations consistently shows a positive bias. This reduces the probability that the experiments are too dry. The second argument is supported by the fact that the relationship between SMB and snow accumulation of Mernild et al. (2017) does not coincide with that of Schaefer et al. (2013). The former shows more positive SMB values for the same accumulation (see Fig. 6). Let us assume for the sake of simplicity that there is a constant offset of $-0.84$ m w.e. yr$^{-1}$ according to the difference between the value provided by Mernild et al. (2017) and the linear approximation. The corrected SMB estimates of the OPM$_{0.45}$ experiment would be shifted towards the range of $-0.46 \pm 0.48$ m w.e. yr$^{-1}$ and therefore be more positive than the geodetic mass balance. The corresponding mass loss by calving would be finally on the order of $-0.44 \pm 0.55$ m w.e. yr$^{-1}$ ($-2.27 \pm 2.84$ km$^{-3}$ yr$^{-1}$). This

is a pure thought experiment, and the numbers can only serve as orders of magnitude.

Furthermore, an invariant and homogeneous liquid-to-solid-precipitation ratio and a universal relationship between annual precipitation sums and SMB has been assumed. Recently published studies indicate that the solid-to-liquid-precipitation ratio varies locally (Bravo et al., 2019). Together with the snowdrift effect, which is also not considered here, this leads to large uncertainties in the mass change estimates (e.g. Sauter et al., 2013). However, this analysis clearly shows how sensitive the estimation of SMB and calving rates react to precipitation uncertainties.

### 4.3 Constrains of the hydrological cycle on the SMB

Given the strong link between the glacier SMB and the local hydrological cycle, the long-term SMB evolution scales with the strength of the WVF, which is, in turn projected to increase in a warming climate. The WVF sensitivity along Patagonia's western coast ($\sim 50°$ S) is on the order of $\sim 15\,\%\,K^{-1}$ ($\sim 3\,\%$ per decade) as a result of the strengthening of the westerlies ($\sim 20\,\%\,K^{-1}$) and increase in IWV ($\sim 5\,\%\,K^{-1}$) south of $45°$ S. The latter is weaker than the change in global-mean IWV which scales according to the Clausius–Clapeyron relation ($7\,\%\,K^{-1}$) but is consistent with the assumption that increased latent heat flux is compensated by the sensible heat flux (Held et al., 2006; Schneider et al., 2010). The observed zonal-wind trend is associated with a bias towards a more positive Southern Annular Mode (Garreaud et al., 2013; Marshall et al., 2017; Thompson and Solomon, 2002).

The change of the WVF leads to stronger moisture flux convergence along the coastal zone west of the Andes main ridge. Ignoring the fact that the solid–liquid ratio changes, which appears to be a reasonable assumption, since temperature changes in the lower troposphere are negligible ($\sim 0.01$ K per decade), a mean mass gain of $0.57\pm0.06$ m w.e. per degree warming ($0.11\pm0.02$ m w.e. per decade) is expected over the SPI. This rate is consistent with other studies (Mernild et al., 2017). Thus, although the precipitation values presented here indicate that the present-day SMB of the Patagonian Icefields is likely not as positive as suggested by previous studies, the SMB can be expected to show an increasing trend under continued warming conditions.

### 4.4 Limitations and nonlinearities

Given the linear nature of the approach used, the knowledge gained must be critically assessed and is only valid under certain conditions. This linear assumption requires a stably stratified atmospheric flow, more precisely one that is given by a positive moist buoyancy frequency. During the study period from 2010 to 2016, the condition was fulfilled in more than 99 % of all days. As a part of this assump-

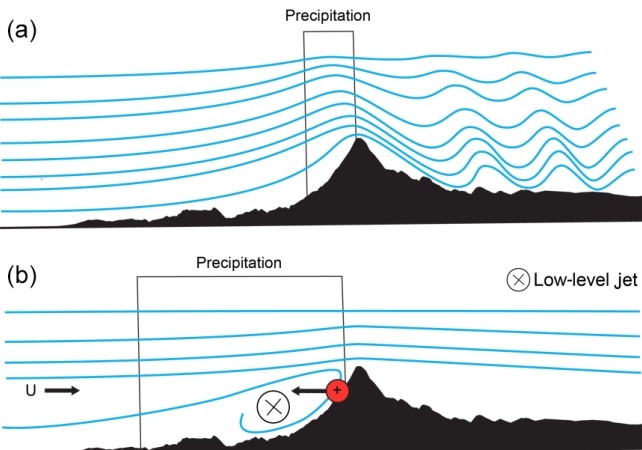

**Figure 7.** Schematic illustration of the interaction between the atmospheric air flow and the Andes. **(a)** Linear mountain flow response ($\hat{H} < 1$) leads to strong uplift and precipitation along the western slopes. **(b)** The air flow is blocked by the topography ($\hat{H} \geq 1$), and the resulting pressure gradient (indicated by the red circle) at the western slope slows down the upstream flow. The imbalance between the large-scale pressure gradient and Coriolis force leads to a northerly low-level jet, which reduces and shifts the uplift motions upstream. This mechanism enhances precipitation in the Precordillera range, while reducing precipitation at the western slopes of the Andes.

tion a linear mountain flow response is required to guarantee that the airflow crosses the mountain range. To ensure a linear-flow regime, the non-dimensional mountain height $\hat{H} = (N_m h_m)/U$ must be smaller than one, where $h_m$ (m) is the mean barrier height. Assuming a mean $h_m = 2200$ m, the conditions ($\hat{H} < 1$) is fulfilled in $> 82\,\%$ of all considered cases (see Fig. S5). In the remaining cases ($\hat{H} \geq 1$), the Andes block the atmospheric flow, and a northerly low-level barrier jet forms along the western slope, parallel to the main ridge (Barrett et al., 2009; Falvey and Garreaud, 2007; Garreaud and Muñoz, 2005; Viale and Garreaud, 2015; see Fig. 7). The low-level jet constitutes an effective barrier to the flow that extends upwind, greatly reducing the uplift motions and thus the condensation of water vapour along the western slopes. The shift in the vertical uplift enhances precipitation upstream of the Andes, while reducing precipitation at the slopes. The effect of blocking is clearly evident in the precipitation fields of high-resolution (500 m) atmospheric simulations of single events using the Weather Research and Forecast (WRF) model (see Fig. 8 and Table S4). Two water-vapour-rich events were chosen to illustrate the influence of the flow regime on the spatial distribution of precipitation. While the linear-flow regime has a pronounced precipitation maximum on the slopes, flow blocking shifts the precipitation far upstream (600–700 km), leading to a more homogeneous pattern.

Upstream precipitation can be further enhanced by microphysical processes such as the seeder–feeder mechanism

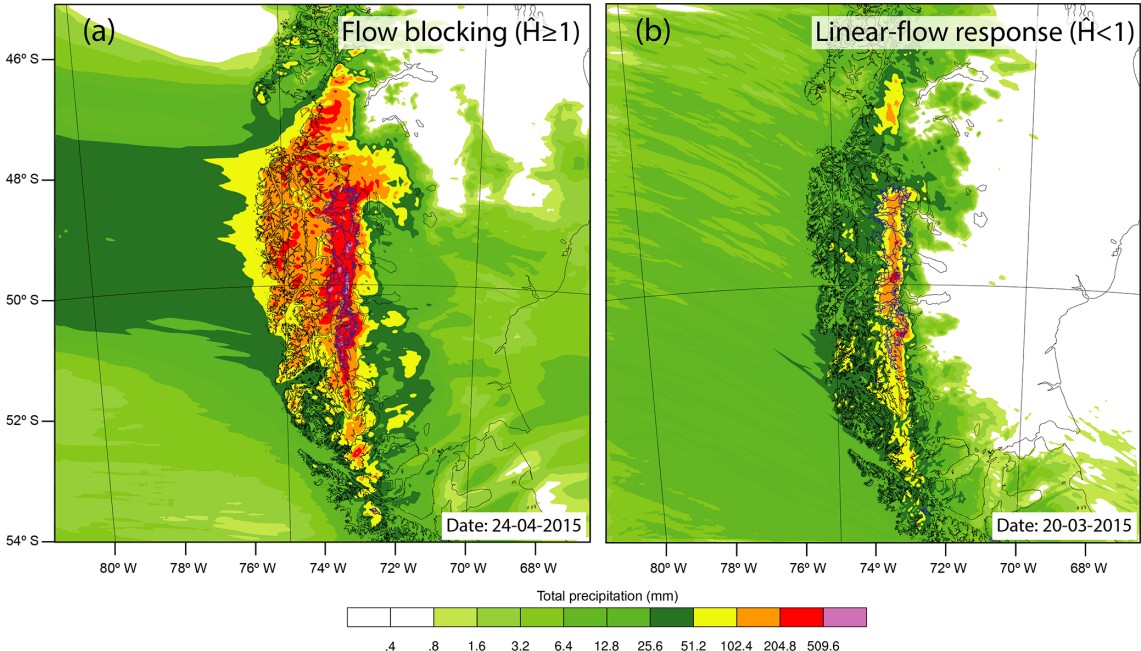

**Figure 8.** Total precipitation sums (3 d) over the SPI and NPI from WRF for different flow regimes. **(a)** Nonlinear-flow response with enhanced precipitation in the Precordillera range and **(b)** linear-flow response with strong localized precipitation along the western slopes of the Andes.

and rapid warm-air autoconversion. Studies have shown that these processes can lead to higher rain accumulations upstream when fronts and embedded atmospheric rivers intersect the western coast of central Chile (Garreaud et al., 2016; Massmann et al., 2017; Viale et al., 2013; Viale and Garreaud, 2015). The lifting of moist air masses upstream produces mid-tropospheric stratiform clouds (seeder), which can be strong enough to produce snow or graupel aloft and light precipitation in the pre-frontal region. If the frontal system is slowed down by blocking, low-level convergence enhances in the area of the narrow cold-frontal rainband and fuels the updrafts. The enhanced updrafts facilitate the development of low-level clouds by collision coalescence between supercooled droplets. When the narrow cold frontal rainband propagates further east, it triggers the seeder–feeder mechanism, and low-tropospheric clouds are seeded by the precipitation that is formed by mid-tropospheric clouds aloft. The associated rapid transformation of cloud water into hydrometeors and increased hydrometeor sizes are absent in the approach presented. Here, the process is treated simplistically by the choice of short timescales and by constraining the synoptic-scale uplift (background precipitation). This solution most likely leads to (i) an overestimation of precipitation on the western slopes of the SPI and (ii) an underestimation of precipitation in the Precordillera zone but (iii) satisfies the given DR constraint. Compliance with the DR criterion is the necessary condition to verify the plausibility of precipitation estimates.

## 5 Conclusion

The present study has shown on the basis of simple physical arguments and a linear model that it is very unlikely that the moisture flux from the Pacific will be sufficient to sustain the reported extreme mean precipitation amounts for Patagonia. While the approaches and assumptions employed in this study contain substantial uncertainties, precipitation estimates using other parameter combinations fall within the range between the two proposed scenarios. Hence, this study offers a plausible range of precipitation estimates based on clearly defined assumptions: (i) the orographically induced precipitation is proportional to the incoming WVF, (ii) the terrain-forced uplift and condensation of moist air masses is assumed to be the dominant precipitation formation process in central Patagonia, and (iii) the atmospheric drying ratio (DR) derived from observed isotope data is a valid measure for the cross-mountain fractionation of the WVF. According to these assumptions, the icefield-wide precipitation averages are likely to fall within $5.06 \pm 0.51$ and $5.99 \pm 0.59$ m w.e. $yr^{-1}$ on the SPI and $5.38 \pm 0.59$ and $6.09 \pm 0.64$ m w.e. $yr^{-1}$ on the NPI. The values within these ranges are about 40 %–65 % lower than previously assumed. Extreme precipitation in wind-exposed regions is in the range of $11.58 \pm 0.98$ m $yr^{-1}$, up to 60 % lower than estimated by other numerical studies (Lenaerts et al., 2014; Schaefer et al., 2013, 2015). It should also be noted that processes such as snowdrift and nonlinear effects have not been taken into account, so the actual accumulation rates are probably still

below these estimates. This result makes it very unlikely that Patagonia is the wettest place on Earth. More importantly, the drier hydroclimatic condition represents a major constraint for the Patagonian Icefields and reduces the precipitation contribution to the glacier mass balance. The missing contribution is evident in the surface mass balance. According to the results, the average SMB (2010–2016) was between $0.56 \pm 0.45$ m w.e. yr$^{-1}$ ($7.82 \pm 6.28$ km$^{-3}$ yr$^{-1}$) and $-0.14 \pm 0.39$ m w.e. yr$^{-1}$ ($-1.95 \pm 5.45$ km$^{-3}$ yr$^{-1}$) on the SPI in the last decades. The mass loss due to calving ranging between $-1.5 \pm 0.64$ m w.e. yr$^{-1}$ ($-20.95 \pm 8.94$ km$^{-3}$ yr$^{-1}$) and $-0.8 \pm 0.58$ m w.e. yr$^{-1}$ ($-11.18 \pm 8.10$ km$^{-3}$ yr$^{-1}$). On the NPI the SMB was more negative with $-0.72 \pm 0.52$ m w.e. yr$^{-1}$ ($-3.72 \pm 2.68$ km$^{-3}$ yr$^{-1}$) and $-1.30 \pm 0.48$ m w.e. yr$^{-1}$ ($-6.72 \pm 2.48$ km$^{-3}$ yr$^{-1}$). The calving flux was estimated to be on the order of $-0.44 \pm 0.55$ m w.e. yr$^{-1}$ ($-2.27 \pm 2.84$ km$^{-3}$ yr$^{-1}$). However, this number is very uncertain. Over the long term, the regional precipitation is likely to increase by $\sim 15\%$ per degree warming ($\sim 3\%$ per decade) in response to stronger moisture flux. Most of the change is related to a strengthening of the westerlies ($\sim 20\%$ K$^{-1}$), while only a minor contribution comes from an increase in IWV ($\sim 5\%$ K$^{-1}$). Assuming that the liquid-to-solid-precipitation ratio and the relationship between annual precipitation sum and SMB are universal and valid for the next decades, the WVF changes would result in a glacier surface mass gain of about $0.57 \pm 0.06$ m w.e. per degree warming on the SPI. This positive trend contradicts the recently published geodetic mass balance observations (Malz et al., 2018) which detected quick glacier recessions in these regions. The observed retreat is significantly stronger than the gain in ice mass, implying that the ice mass budget is partially decoupled from the climate signal and primarily caused by dynamic adjustments of tidewater and lake-calving glaciers. The pronounced dynamic glacier response emphasizes that ice dynamic processes need to be given more prominence in order to quantify the response of the Patagonian glaciers to climate change and their contribution to future sea-level rise. While the change in ice masses is a vivid example of the response to reduced precipitation, it also opens new perspectives for future studies on environmental change in Patagonia and can also help reduce uncertainties in the quantification of other precipitation-driven environmental phenomena.

*Data availability.* Data were obtained from the European Centre for Medium-Range Forecast (ECMWF, 2018), National Aeronautics and Space Administration (NASA; NASA/CIGRA, 2018), and Integrated Global Radiosonde Archive (IGRA; NOAA, 2017a) from the National Centers for Environmental Information (NCEI). SSM/I and SSMIS data are produced by Remote Sensing Systems. Data are available at http://www.remss.com/missions/ssmi (NOAA, 2017b).

*Supplement.* The supplement related to this article is available online at: https://doi.org/10.5194/hess-24-1-2020-supplement.

*Competing interests.* The author declares that there is no conflict of interest.

*Acknowledgements.* I would like to thank Manuel Gebetsberger, Emily Collier, Thomas Mölg and Nicolas Cullen for the discussion and support. The simulations were calculated on the High-Performance Cluster (HPC) at the Regional Computation Center (RRZE) of the University of Erlangen-Nürnberg.

*Financial support.* This research has been supported by the German Research Foundation (DFG) (grant no. SA 2339/4-1).

*Review statement.* This paper was edited by Bettina Schaefli and reviewed by Claudio Bravo and Marius Schaefer.

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

**Remarks from the typesetter**

TS1    According to our standards, changes like this must first be approved by the editor at this stage. Please provide a detailed explanation for those changes that can be forwarded to the editor. Please note that this entire process will be available online after publication. Upon approval, we will make the appropriate changes. Thank you for your understanding.