# Peer review of "Revisiting extreme precipitation amounts over southern South America and implications for the Patagonian Icefields"

_Hydrology and Earth System Sciences, 2019_

## Referee Comment (RC1) · Claudio Bravo (Referee) · 19 Jul 2019

Title: Revisiting extreme precipitation amounts over southern South America and implications for the Patagonian Icefields

PAPER SUMMARY AND RECOMMENDATION

Sauter presents an evaluation of the precipitation magnitude across southern South America. The emphasis is given in the implications to the surface mass balance (SMB) of both Icefields. New data is mainly compared with previous SMB models. Based on the data obtained from an Orographic Precipitation Model, the main conclusions are that Patagonia is not the wettest place on Earth and that the previous SMB overestimates the accumulation. The scientific significance is high regarding the lack of information (and observations) of the actual magnitude of the precipitation on the Patagonian Icefields. The scope of the research is clear and prove (based on the used datasets and assumptions), that higher amount of precipitation previously reported are not sustained by the moisture flux over the region, although this conclusion, of course, will continue to be subject of discussion in the future. The manuscript is within the scope of HESS, it is in general well-written. The reason for the study must be well justified in the Introduction. The Methodology section needs to clarify and add some explanations, as well as some dataset and experiments details used for validation/comparison. The Results/Discussion section needs to be strengthened. I think this paper provides interesting data and results to the glaciological and hydro-climatological communities. In my opinion, it must be considered for publication after solving/clarifying some issues (see Major comments) and a careful revision of the text (see Minor comments).

MAJOR COMMENTS

1. Introduction

As one of the main topics is the implication of the precipitation on the mass balance on the Icefields, a detailed review of the previous estimations is necessary, considering also the title of the Manuscript. This is barely mentioned, but a comprehensive revision will be helpful for the reader and also will give a strong justification to the present work, mainly related to the fact that besides all the past efforts, still high uncertainties exist in the precipitation magnitude and/or accumulation rate on the Patagonia Icefields.

In this section, it is mentioned the importance of the atmospheric rivers as a source of moisture on Patagonia (first paragraph), but a previous work (Langhamer et al., 2019) indicated that the main source of moisture for the South Patagonian Icefield is the Pacific Ocean between 30° and 60° S. Please, clarify this.

**2. Soundings**

It seems that the soundings are used to validate and corrected the WVF from ERA-Interim. Although the information about the WFV trend for both soundings is interesting, it is not used in the analysis. A Figure showing the comparison of the observations with the ERA-Interim data will be useful to understand the bias correction (Page 5, Lines 15-20). Please note that the SSMIS data is not mentioned in the Methodology section as a dataset used for comparison.

**3. Spatial differences**

It is well established that there are extreme climate gradients on the region, and the results of this work also provide evidence of this, however, in terms of SMB just a mean value is used for each Icefield (NPI and SPI). The author must consider the spatial differences and/or the heterogeneous response of theses glaciers determined by the geodetic method (e.g. Malz et al., 2018; Jaber et al., 2019). Overall, the discussion is related to the glacier shrinkage, despite that, some glaciers show positive (e.g. Pio XI) and stable mass balance.

**4. Mass balance uncertainty**

The approach to calculating the mass balance is simple but useful to demonstrate the implication of the overestimation of the accumulation rate. However, it must be mentioned other source of uncertainty on SMB estimations. Recent work indicates that the method chosen to estimate the ratio of solid precipitation is also a source of uncertainty in accumulation estimations (Bravo et al., 2019). This even could lead to lower accumulation rate than those mentioned in the manuscript as the method used is quite simplistic.

**5. Limitations and nonlinearities**

This section presents a comprehensive analysis of the main source of uncertainty, related to the linear nature of the model. This is quite interesting and necessary in a work

of this nature, and the last three lines of this section (Page 8, Lines 11-14) is a perfect summary of this section. However, this analysis should be useful if quantification of the uncertainty is given. This quantification could be obtained from the WRF experiment, which also needs to be mentioned in the Methodology section. This section also uses three of a total of seven Figures to explain the limitations, please check if is really necessary the three Figures.

6. Consistency on percentage

Please clarify the percentage of overestimation regarding previous works. For instance, it is mentioned in Page 6 Lines 15-16 that the maximum precipitation (11.58 +-0.98 m yr-1) represents a reduction of 60% compared to other numerical studies, while in the Conclusions section (Page 8, Lines 21-22) this same value represents a reduction of 30-50%. Please also check the other percentages given.

MINOR COMMENTS:

Please perform a careful revision of the References.

Define in the text acronyms not defined: GPM, SSMIS, WRF, OPM

Just for consistency, use the same units. In the text, the units are in some sections as m yr-1, while other sections and in Table 1 the units are m w.e. yr-1. This could produce some confusion. Please also note that the work of Bravo et al. (2019) is snow accumulation and no total precipitation.

I suggest to include some of the equations that are explained in the Methodology sections in Page 4 lines 3-10.

On Page 2, most of the last paragraph should be on the Methodology section (assumptions and conditions).

Page/Lines:

3/1: Define "DR"

3/11 "DR-scaling"

3/14: "Langhamer et al. (2018)" is not listed in References

3/ 17: Define "GPM"

3/24: Indicate the vertical precipitation gradient of Schaefer et al. (2013)

3/26-27: "orographic precipitation model (OPM)"

3/28: "Garreaud et al., 2016" is not listed in References

3/33: "the decay"

4/ 11: "derived"

4/13: "remarkably"

4/20: "delivers"

4/27-28: Add period of the average WVF given.

4/29: Add reference to this sentence.

5/12: Define SSMIS

5/17: "suggests"

6/1-2: "Dirección General de Aguas"

7/10: "Southern Annular Mode (SAM)" or "Antarctic Oscillation (AAO)"

7/18: "Given" instead of "In view of"

7/19: Delete "necessary"

8/16: "based on" instead of "on the basis of"

8/20: "based on"

8/26: Delete "clearly"

8/29: "Assuming that changes in the orographically induced precipitation are..."

8/31: "per degree of warming"

9/3: Delete "in order"

Figure 1: "show"

Figure 2: Add units to Y-axis

Figure 4: Reference to SPI is "Schaefer et al. (2015)" no "Schaefer et al. (2013)"

Figure 4: Caption indicate that the period is "1975-2000", however, the number of blue points (annual values) is greater than this period covers. Correct or clarify this. Maybe "1975-2011"?

Figure 4: Complete last line of the caption: "Schaefer et al.."

Table S2: Please indicate which Stations are leeward and which are windward. This will be helpful to fully understand the manuscript on Page 6, Lines 1-4.

---

## Referee Comment (RC2) · Marius Schaefer (Referee) · 6 Sep 2019

The comment was uploaded in the form of a supplement:
https://www.hydrol-earth-syst-sci-discuss.net/hess-2019-225/hess-2019-225-RC2-supplement.pdf

---

## Editor Comment (EC1) · Bettina Schaefli (Editor) · 13 Sep 2019

Both reviewers agree that this manuscript is an important contribution but that it requires some methodological clarifications and improvement of the presentation of the results and discussion. I invite the author to answer these methdological and content / structure comments here in the public discussion before preparing the revised version.

---

## Author Comment (AC1) · 9 Oct 2019

I thank the reviewer for the constructive comments and suggestions. My response to the review can be found in the attached document.

**R**: Referee's comment
**A**: Author's response
**C**: Proposed changes in the manuscript
* * *
**Summary**

**R:** Sauter presents an evaluation of the precipitation magnitude across southern South America. The emphasis is given in the implications to the surface mass balance(SMB) of both Icefields. New data is mainly compared with previous SMB models. Based on the data obtained from an Orographic Precipitation Model, the main conclusions are that Patagonia is not the wettest place on Earth and that the previous SMB overestimates the accumulation. The scientific significance is high regarding the lack of information (and observations) of the actual magnitude of the precipitation on the Patagonian Icefields. The scope of the research is clear and prove (based on the used datasets and assumptions), that higher amount of precipitation previously reported are not sustained by the moisture flux over the region, although this conclusion, of course, will continue to be subject of discussion in the future. The manuscript is within the scope of HESS, it is in general well-written. The reason for the study must be well justified in the Introduction. The Methodology section needs to clarify and add some explanations, as well as some dataset and experiments details used for validation/comparison. The Results/Discussion section needs to be strengthened. I think this paper provides interesting data and results to the glaciological and hydro-climatological communities. In my opinion, it must be considered for publication after solving/clarifying some issues (see Major comments) and a careful revision of the text(see Minor comments).

**A:** I appreciate very much the overall positive evaluation of the script and will follow the suggestions of the expert in the respective sections. I also agree with the expert's opinion that the discussion about the precipitation in Patagonia is not yet finished and that further relevant studies will be necessary in the future to close the knowledge gap and open issues. I am convinced that this work is a step in the right direction.
* * *
**Introduction**

**R**: As one of the main topics is the implication of the precipitation on the mass balance on the Icefields, a detailed review of the previous estimations is necessary, considering also the title of the Manuscript. This is barely mentioned, but a comprehensive revision will be helpful for the reader and also will give a strong justification to the present work, mainly related to the fact that besides all the past efforts, still high uncertainties exist in the precipitation magnitude and/or accumulation rate on the Patagonia Icefields.

**A**: In the present version, only the range of previous mass balance studies with the corresponding literature sources was deliberately mentioned. Most of the literature sources mentioned here summarize the mass balance studies published so far. Since no scientific added value is generated by a new summary, only the previous studies were referred to here. In order to emphasize the importance of the study further I will follow the suggestions of the expert gladly and will work out more precisely the necessity of the study in view of existing work.

**C**: The last sentences of the first paragraph of the introduction (p2 l8-15) will be revised with respect to existing studies.

**R**: In this section, it is mentioned the importance of the atmospheric rivers as a source of moisture on Patagonia (first paragraph), but a previous work (Langhamer et al., 2019) indicated that the main source of moisture for the South Patagonian Icefield is the Pacific Ocean between 30◦and 60◦S. Please, clarify this.

**A**: Atmospheric rivers from the Pacific always lead to a strong increase in water vapour transport and consequently to the strongest precipitation events in Patagonia. These water vapor-rich outbursts occur in the tropical Pacific as a result of geostrophic turbulence. The statement in our paper (Langhamer et al., 2018) that the dominant moisture source in the South Pacific is between 80-160ºW and 30-60ºS does not contradict this statement but confirms the observation. Furthermore, looking at the identified moisture source patterns of Langhamer et al (2018) one can see that this also corresponds to the statistically mean tracks of atmospheric rivers.

**C**: At the beginning of the introduction (p1 l24-27) I will refer explicitly to the study by Langhamer et al. (2018) and its results.
* * *
**Soundings**

**R**: It seems that the soundings are used to validate and corrected the WVF from ERA-Interim. Although the information about the WFV trend for both soundings is interesting, it is not used in the analysis. A Figure showing the comparison of the observations with the ERA-Interim data will be useful to understand the bias correction (Page 5, Lines15-20). Please note that the SSMIS data is not mentioned in the Methodology section as a dataset used for comparison.

**A**: Yes, the radiosondings were used in this study exclusively to evaluate the ERA interim data. A detailed analysis of the trends was not in the spotlight here. Nevertheless, I fully agree with the reviewer regarding the relevance of these data. The WVF time series show very exciting phenomena, such as the influence of atmospheric modes or the presence of decadal signals (water vapor flow) in Punta Arenas while a continuous trend is observed in Puerto Montt. This raises the pressing question which processes can lead to this very different response and to what extent this is related to a changing climate. These exciting questions might open up new perspectives that need to be addressed in detail in follow-up studies and would go beyond the scope of this work.

**C**: In Figure 2 the time series of the WVF anomaly of the ERA interim were also shown in the first version. For the sake of clarity, the time series were taken out again and the statistical differences (bias, trend, etc.) were presented in the text. If desired, I will add the ERA interim anomalies again in Fig. 2. Many thanks also for the note that the SSMIS dataset was not introduced. I will introduce the dataset accordingly.
* * *
**Spatial differences**

**R**: It is well established that there are extreme climate gradients on the region, and the results of this work also provide evidence of this, however, in terms of SMB just a mean value is used for each Icefield (NPI and SPI). The author must consider the spatial differences and/or the heterogeneous response of theses glaciers determined by the geodetic method (e.g. Malz et al., 2018; Jaber et al., 2019). Overall, the discussion is related to the glacier shrinkage, despite that, some glaciers show positive (e.g. Pio XI) and stable mass balance.

**A**: The extreme climate gradients are probably one of the strongest worldwide and probably not only in terms of the drying ratio, as was shown by the reviewer in his relevant study (Bravo et al., 2019). These inevitably lead to a strong west-east differentiation of the surface mass balance and, in interaction with the dynamic effects, to very heterogeneous responses. In our study (Braun et al., 2019) we were able to show by means of TandemX satellite data that the dynamic adjustment of large glaciers plays a decisive role. Evidence has been provided that ice dynamic probably leads to temporary stable mass balances of some glaciers. Unfortunately, such a differentiated analysis is not possible within the

approach of this study. This is mainly due to the fact that the SMB is derived from the simplified SMB-accumulation relationship derived from the work of Schaefer et al. (2015) (see Fig. 4). This approach can only be applied to the overall mass balance and is not valid for a differentiated consideration. However, Table 1 can be extended accordingly to emphasize the strong west-east differentiation. Here new columns can be inserted which show the mean precipitation values for individual height zones on the west or east side.

**C**: I will extend Table 1 and point out again in Section 4 that the approach does not allow differentiation in the mass balance.
* * *
**Mass balance uncertainty**

**R**: The approach to calculating the mass balance is simple but useful to demonstrate the implication of the overestimation of the accumulation rate. However, it must be mentioned other source of uncertainty on SMB estimations. Recent work indicates that the method chosen to estimate the ratio of solid precipitation is also a source of uncertainty in accumulation estimations (Bravo et al., 2019). This even could lead to lower accumulation rate than those mentioned in the manuscript as the method used is quite simplistic.

**A**: This is absolutely correct and the other sources of uncertainty have not yet been sufficiently addressed.

**C**: Other sources of uncertainty, such as uncertainty in the solid-liquid relationship, are explored in more detail in the discussion at the end of Section 3.3.
* * *
**Limitations and nonlinearities**

**R**: This section presents a comprehensive analysis of the main source of uncertainty, related to the linear nature of the model. This is quite interesting and necessary in a work of this nature, and the last three lines of this section (Page 8, Lines 11-14) is a perfect summary of this section. However, this analysis should be useful if quantification of the uncertainty is given. This quantification could be obtained from the WRF experiment, which also needs to be mentioned in the Methodology section. This section also uses three of a totals of seven Figures to explain the limitations, please check if is really necessary the three Figures.

**A**: A detailed quantification of the uncertainties can hardly be carried out due to the sparse observations. In section 3.2 (and table S2), the mean deviations of the simulations from the observations (p6 l2-5) were discussed as far as possible.

"…*Comparison with in-situ observations from the Dirección de  General de Aguas (DGA, Chile) indicates that the model slightly overestimates precipitation on the leeward side by 0.29±0.37 m yr$^{-1}$(see Table S2). Greater deviations (1.07±1.30 m yr$^{-1}$) occur at the stations on the west side which are located at the foot of the Patagonian Icefields. The overestimation is the result of the rapid increase in model terrain elevation and the absence of nonlinear processes in the linear model (see Sec. 3.4)…*"

A comparison with the WRF simulations is not suitable for the quantification of the uncertainty for two main reasons: (i) WRF simulations were only performed for single events to analyse the effect of atmospheric blocking on the precipitation distribution, and (ii) the simulations themselves cannot be considered as 'reality' and are themselves subject to very high uncertainties. For the mentioned reasons, a detailed quantification is only conditionally possible.

The three figures have been chosen so that the limitation of the approach and the nonlinear nature of the processes are evident. Figure 5 shows how often such atmospheric blocking events occur and how often the approach has to be challenged. The subsequent figure shows the underlying processes and theoretical effect on precipitation that may not be familiar to every reader. Figure 6 then finally provides numerical evidence for the effects.

**C**: If the readability of the article is impaired by the number of figures or the number exceeds the allowable number, I would suggest not to use Figure 5.
* * *
**Consistency on percentages**

**R**: Please clarify the percentage of overestimation regarding previous works. For instance, it is mentioned in Page 6 Lines 15-16 that the maximum precipitation (11.58 +-0.98 myr-1) represents a reduction of 60% compared to other numerical studies, while in the Conclusions section (Page 8, Lines 21-22) this same value represents a reduction of 30-50%. Please also check the other percentages given.

**A**: Thank you for pointing that out. This is an error and the correct information should be read 'up to 60%'.

**C**: I will check the consistency of the percentages throughout the document and adjust them if necessary.
* * *
**Minor comments**

**A:** I will review and correct all the comments made by the reviewer.

---

## Author Comment (AC2) · 9 Oct 2019

I thank the reviewer for the constructive comments and suggestions. My response to the review can be found in the attached document.

**R**: Referee's comment
**A**: Author's response
**C**: Proposed changes in the manuscript
* * *
**General comments**

**R**: Generally I think it is a study which has the potential to make an important contribution to our understanding of precipitation patterns in the study region. However, giving that the results of this study are quite different from previous numerical studies, I am missing are more dedicated search for the reasons of this differences in the discussion section ( I am giving some ideas in the specific comments). Furthermore I think that an analysis of the implications of this study for the surface mass balance of the Northern Patagonia Icefield (NPI) would add very much. Here the results could be validated much better against geodetic glacier mass balances since the losses by calving are much better constraint than for SPI (only one tidewater calving glacier on the NPI). Also readability of the manuscript (especially for not climate modelers) could be easily improved. Please find more detailed comments below.

**A**: Many thanks for the fundamentally positive assessment of the study. I will implement the constructive remarks and comments conscientiously and make the manuscript more accessible to a wider readership. The extension of the study to the NPI would certainly be interesting, but the linear relationship between precipitation and SMB does not necessarily apply to the NPI. Therefore, I would prefer to omit the NPI analysis.
* * *
**Specific comments**

**R**: Line11: "volume loss of the Patagonian Icefields, for example, contradicts the reported positive surface mass balances" there is no contradiction if the difference can be attributed to calving fluxes (or other mass losses).Example Antarctica: positive SMB, negative overall MB.

**A**: That's correct.

**C**: I'll re-write the sentence, for example:

"… The Patagonian Icefields, for example, are one of the largest contributors to sea-level rise outside the polar regions, and robust hydroclimatic projections, in particular estimates of precipitation, are needed to understand and quantify current and future changes. The reported projections of precipitation from numerical modelling studies tend to overestimate those from in-situ determinations and the plausibility of these numbers have never been carefully scrutinised, despite the significance of this topic to our understanding of observed environmental changes. …"

**R**: Line11-14: Reformulate: you are using a model in this study (and not only a simple physical argument) and get some results. Describe the model briefly: what are the input data and main assumptions.

**A/C**: I will reformulate the text.
* * *
**Introduction**

**R**: Line 12/13: up to 30 m w.e. yr -1 are suspected at isolated locations (Lenaerts et al., 2014; Mernild et al., 2017; Schaefer et al., 2013, 2015;Schwikowski et al., 2006). Revise citations! In Schaefer et al.,

2013, 2015 no precipitation of up to 30 m w.e. yr$^{-1}$ are suspected (I think only Lenaerts et al., 2014 is mentioning this value!)

**A/C**: In fact, the figure of 30 m w.e. yr$^{-1}$ was mentioned only by Lenaert and the estimates of the other studies are lower. For this reason the formulation 'up to 30 m' was used and not 'all studies show values of 30 m'. I confess that the formulation can be somewhat confusing and will correct this.

**R**: Line 29-34: to improve readability, I would recommend to leave the assumption ii) shorter. 1-2 sentences. You can explain the technical details in the method section.

**A**: In order to increase the readability I will shorten the second assumption slightly.

**C**: I will move the following sentences to the method part: "This criterion requires a stably stratified atmospheric flow, more precisely given by a positive moist buoyancy frequency. During the study period from2010 to 2016, the condition was fulfilled in more than 99% of all days. … The linearity requirement was met in 82% of the cases (see Sec. 3.4)".

**R**: Line 7: Replace "and critically reviewed (Section 3.4) " by "limitations of the linear precipitation model are discussed in Section 3.4"

**A/C**: Will be done.
* * *
**Methodology**

**R**: To improve readability I recommend to divide the methods section in three subsections:
2.1 DR-scaling using a constant precipitation gradient
2.2 DR-scaling using the linear precipitation model
2.3 Examination of non-linearities using the Regional Climate Model WRF (new section)

**A/C**: In order to make the chapter clearer, I will divide the section into the following subchapters:
2.1 DR-scaling
2.2 Linear orographic model
2.3 Numerical Simulations with the Weather Research and Forecast (WRF) Model

**R**: p3 l16: "by optimizing (Newton-Raphson algorithm) the vertical precipitation gradient" How does this optimizing work? Describe in one sentence and give a reference for further reading.

**A**: The Newton-Raphson algorithm is a standard optimization method and in my opinion only needs a corresponding reference.

**C**: I will add a corresponding reference.

**R**: p3 l17: "determined from the GPM measurement" : determined : how? Or do you mean taken from? Acronym GPM not explained!

**A**: The abbreviation was indeed introduced only in the caption of Figure 1 and not in the text.

**C**: I will introduce the abbreviation 'GPM' on p3 l17 and replace 'determined' with 'taken from'.

**R**: p3 l18/19: "The optimization resulted in a vertical precipitation gradient of 0.00052 m m$^{-1}$ (~0.02% m$^{-1}$), which represents a slightly smaller lapse rate than previously reported (Schaefer et al., 2013)" In Schaefer et al. a precipitation of 5% per 100m was employed. That is more than double the one you used. You could create another scenario using a lapse rate of 5% per 100m!

**A**: The numbers mentioned in the text still refer to old simulations and are not correct. The optimized lapse rate is $5.29*10^{-2}$ % m$^{-1}$, and thus 5.3 % per 100 m.

**C**: I will change the numbers in the text accordingly.

**R**: p3 l27: "many processes" : name some (or all)!

**A/C**: Will be done.

**R**: p3 l31: "it solves two steady-state advection equations" : show the equations or give reference

**A/C**: I'll give the references to the corresponding papers.

**R**: p3 l32: "conversion from cold water to hydrometeors" : cold water IS a hydrometeor? Do you mean cold vapor?

**A/C**: The sentence must read as 'cloud water' and not 'cold water'. Will be corrected.

**R**: p4 l2: "and properties" : properties of what? Which properties do you mean?

**A/C**: The uniform flow characteristics refer to wind speed, atmospheric stability, scale height etc. (i.e. all assumptions made for this model and explained in the following sentences). Only under these assumptions a linear model can be developed at all. I will present this more clearly.

**R**: p4 l3: "five parameters" : could explain each parameter in one sentence?

**A/C**: I will revise the description of the linear model so that the purpose of the parameters is easier to understand.

**R**: p4 l8: "background precipitation is scaled by a constant" : which range of values does this constant take? What happens if you do not force the model to a fixed drying ratio?

**A**: Since the model is based on the linear (mountain wave) theory, the precipitation is extended into a background value and a perturbation (orographic part). Here it is assumed that the background value does not vary at all or only very slowly. It must also be considered that the perturbations have no influence on the background value. Thus the orographic precipitation model can only represent the orographic induced part of the precipitation and not the total precipitation. The synoptically induced precipitation must therefore be added. In the study there is no constant value for the background value but is adjusted by the drying ratio (including white noise). So if no background value is added, only the orographic part is obtained, which leads to unrealistic total precipitation.

**R**: p4 l10: "$C_w$ and $N_m$ are calculated from 6-hourly ERA-Interim fields" how?

**A**: $C_w$ relates the condensation rate to vertical motion in the atmosphere and can be derived from the ratio of the moist adiabatic and atmospheric lapse rates. The effective moist static stability is approximated from the commonly used equation (Fraser et al., 1973; Smith and Barstad, 2004).

**C**: I will clarify in the text how these parameters have been derived.

**R**: p4 l14/15: "produce remarkable similar results" which kind of results? Precipitation fields? You should show that in the results and discussion section!

**A**: Garreaud et al. (2016) has used the WRF model and the linear model to study the orographic precipitation in coastal Southern Chile. They found similarity and high spatial correlations on longer time scales (not daily) between the two models. The findings cannot be directly applied to the study presented, but indicate that the linear model leads to consistent results under realistic setups.
* * *
**Results and Discussion**

**R**: p4 l27/28: "Along the coast ..." to which data refers this sentence? ERA-Interim cells located at the coast or measurements in Puerto Montt or Punta Arenas?

**A/C**: 'Along the coast' refers to the Pacific coast where Puerto Montt is located. I agree, this is confusing and will be re-written.

**R**: p4 l 29/30 Sentence starting with "There is also clear evidence ..." needs citation. Or are you referring to the data in Puerto Montt?

**A/C**: The evidence follows from Fig. 2 which provides the basis for the discussion (paragraph) here. I will add another citation to Fig. 2 here.

**R**: p5 l6 : " The ERA-Interin data ..." : please add these data in Figure 2.

**A/C**: (see also response to reviewer 1). In Figure 2 the time series of the WVF anomaly of the ERA interim were also shown in the first version. For the sake of clarity, the time series were taken out again and the statistical differences (bias, trend, etc.) were presented in the text. If desired, I will add the ERA interim anomalies again in Fig. 2.

**R**: p5 l12: "SSMIS data" : explain! Are these reliable data?

**A/C**: (see also response to reviewee 1). The SSMIS dataset was not introduced at this point, but I will provide more information on the dataset. It is reported that the data is reliable over the ocean but underestimates the vertical integrated water vapor over cold surface such as glaciers.

**R**: p5 l18: "and corrected accordingly" : this means you multiplied the original ERA-interim WVF by 1.1?

**A**: Yes, the WVF has been bias corrected with a factor 1.1

**R**: p5 l22: "Pre-Cordillera region" add (see Figure 1)

**A/C**: Will be done.

**R**: p5 l29 ".. values agree with precipitation estimates from discharge measurements " this statement is not true (see table 1).

**A**: One would have to say is closer to the estimates from the discharge measurements.

**C**: I will remove this remark.

**R**: p5 l32: "windward side" and "leeward slopes" what is the extend of this regions? Can you indicate them in Figure 1 or 3?

**A/C**: The terms refer to all slopes of the Patagonian Icefields facing to or away from the wind. The west-east differentiation is easy to see in Figure 3.

**R**: p5 l33: "The spatial pattern on the plateau is consistent ..." how can a precipitation pattern be consistent with elevation change measurements? Explain! How do you define "the plateau" ? Indicate this region in Figure 3!

**A/C**: The remark by the reviewer is justified because the statement can only valid if hardly any ice dynamic processes take place, which was assumed here for the higher area near the ice divide. This is a very simplistic assumption and not necessarily based on hard facts. But the simulated accumulation patterns are very similar to the observations of the geodetic method. It is clear to me that this statement raises further questions which cannot be answered in the context of this study. I would therefore suggest to discard this sentence.

**R**: p6 l3: "Greater deviations" better say "higher overestimations"

**A/C**: Will be implemented accordingly

**R**: p6 l7: "Shiraiwa et al. is not a presenting simulations but measurements. Please indicate that clearly!

**A/C**: Ok.

**R**: p6 l7 : " The large ensemble spread" : how much is it? You have to show that!

**A**: The ensemble spread is reflected in the uncertainties of the mean values (see e.g. Table 1).

**R**: p6 l9-10: " the responsible mechanisms explaining the significant differences remain unclear." That's a very sad statement for a scientific contribution! At least try! Potential candidates are: higher drying ratios or higher precipitation gradients in other studies. You also ran the regional climate model WRF. What drying ratios and precipitation gradients did you get from this simulations? Also different study period could be a candidate.

**A**: The source of uncertainty in such cases can be very diverse. In addition to the actual errors in the input data, uncertainties in the parameterizations are particularly relevant.  Some microphysical

parametrisation schemes are more 'grauple-friendly' than others which can lead to strong hyrdometeor formation. In addition, a multitude of parameterization combinations can lead to very different results. Each model setup must therefore be examined individually. Errors in the drying ratio are the direct consequence of inadequate process mapping. The own WRF simulations do not provide any further information since no longer time periods were calculated. For single events the DR can deviate strongly from the long-term mean and the isotope measurements. The sources are manifold and can only be speculative in the context of this study.

**C**: Even if it is very speculative, I will revise this paragraph and work out possible sources.

**R**: p6 l26-28: "…, we use the significant linear relation (), between annual precipitation sum and annual SMB derived from data of Schaefer et al. (2015). The relationship is: …

**A/C**: Will be changed accordingly.

**R**: p6 l30/31 "...would result in a mean SMB between 0.56±0.45 m w.e. yr -1 (7.82±6.28 km -3 yr -1, extreme scenario ) and -0.14±0.39 m w.e. yr -1 (-1.95±5.45 km -3 yr -1, realistic scenario ) on theSPI (Fig. 4)

**A/C**: Will be changed accordingly.

**R**: p7 l1-2: "… the mean mass loss due to calving ranges between -1.5±0.64 (-20.95±8.94 km -3 yr -1) and -0.8±0.58 m. w.e. yr -1 (-11.18±8.10 km -3 yr -1 )" Are these values realistic considering that recently calving fluxes of up to 3.81 km³w.e. yr-1 for one single glacier were observed during the study period (2015)?
Citation: Bown F, Rivera A, Pętlicki M,Bravo C, Oberreuter J, Moffat C (2019). Recent ice dynamics and mass balance of Jorge Montt Glacier, Southern Patagonia Icefield. Journal of Glaciology 1–13. https://doi.org/ 10.1017/jog.2019.47

**A**: The values presented here are mean values over the entire period (7 years). Individual events can lead to large mass losses which increase the total mass loss in individual years. The event of 2015 cannot be represented with the approach presented here, since in this study the mean SMB is used together with the geodetic mass balance observations which also represented an integrated value. Any error in the SMB estimate inevitably leads to further uncertainties in the calving fluxes.

**C**: To remind the reader that such individual events are not considered in this study and that individual years can deviate strongly from the values, I will take up the study of Brown et al. (2019) in the discussion.

**R**: More results of the WRF simulation should be presented in the results section: which precipitation gradients, total precipitation values of the icefields and drying ratios do you obtain from these simulations?

**A**: As already mentioned in the previous answers, high-resolution WRF simulations were only calculated for single blocking events. The values derived from a few single events such as precipitation gradients, drying ratio and total precipitation would not be representative.
* * *
**Conclusions**

**R:** p8 l16: " … simple physical arguments ..." again: you are using a numerical model, not a simple physical argument. Better repeat model essentials.

**A**: A linear (analytical) model is not a numerical model in a strict sense. A numerical model generally consists of a model conception, a closed physical system of equations which are solved by a numerical solver. A analytical model has a mathematical closed form solution. DR-scaling in turn is a clear scaling approach by scaling precipitation by water vapor flux.

**C**: I suggest to continue to use the term 'scaling' or 'physical argument' for DR-scaling approach and to use the term 'analytical model' or 'linear model' for the orographic precipitation model.

**R**: p8 l18: ".. other parameter combinations ..." of the model employed here or generally?

**A**: The statement refers to the models used here. Since these are linear models, all parameter combinations that lie between the realistic scenario and the extreme scenario must also must lead to precipitation values that lie in between.

**R**: p8 l20: " … clearly defined assumptions. " add which are: …

**A/C**: Will be done.

**R**: p8 l22: Shiraiwa et al.,2002 is NOT a modeling study!!!

**A/C**: I will make that clear.

**R:** p8 l30/31: "WVF changes would result in a glacier surface mass gain of about 0.57±0.06 m w.e. per degree warming. TAKE CARE HERE!!! The relationship between precipitation and SMB you derived from the data of Schaefer et al. 2015 will not be valid forever. Percentage of solid precipitation will decrease for higher temperatures!

**A/C**: I'm fully aware of that. For this reason I will repeat at this point the statement of page 7 line 12 'Ignoring the fact that the solid-liquid ratio changes' and explicitly point out that this trend can only be correct for short periods (including a reference to the work of Bravo et al. (2019)).

**R**: Line 4 : "While the change in ice masses ..." to which change are you referring to?

**A/C**: This is a general statement suggesting that the change in ice masses is a very illustrative example of how precipitation uncertainties can affect systems.
* * *
**Figures**

**R (Figure 1)**: I am a bit surprised about the many dots! Can you indicate a list of stations (supplementary material) and indicate to which time span the color coding of the measurements corresponds?

**A/C**: Yes, I will add a column with the corresponding time periods to Table S2.

**R (Figure 2)**: As indicated before, I would like to see the closest ERA-interim gridpoint data added for each station.
would prefer very much absolute data instead of anomalies!

**A/C**: See previous comment. I prefer to consider the anomalies to get rid of the seasonality without interfering with the trends. It also helps us to compare the different Puerto Montt with Punta Arenas.

**R (Figure 3)**: Both plots are qualitatively very similar. Could you better compare the linear prec. gradient approach with the numerical model (using the same drying ratio)?

**A/C**: This is not possible since the WRF simulations are event based simulations (see previous comments).

**R (Figure 4)**: revise caption! The red, green and orange dots are the results of this study! Schaefer et al. (2013), Mernild et al. (2016) are indicated as red circles.

**A/C**: I will revise the figure caption.

**R (Figure 4)**: I not very necessarily I find. Better add a nice figure about the model validation at weather Stations.

**A/C**: will remove the figure (see also comment of Reviewer 1).

**R (Table 1)**: Lenearts et al. (2013) are indicating mean precipitation estimates in gigatons. You can calculated specific values from that. Also revise the max values ( I think 30 m was obtained at SPI).

**A/C**: Thank you for pointing that out. I will check it out.
* * *
**Supplementary material**

**R**: A nice graphical representation of Table S2 should be added to the main part of the manuscript. Results of all scenarios should be validated against station data. Perhaps you could realize two scatterplots: one for the windward side and one for the leeward side? Add observation period to the table (or new table, see comment Figure1!).

**A/C**: I will extend table S2 with additional columns that indicate the observation period and the results of the extreme scenario and DR scaling. If the scatterplots are meaningful I will gladly add them.

---

## Author Response (AR1)

[revised manuscript text omitted]

| Field Coc              | le Changed                                                 |
|------------------------|------------------------------------------------------------|
| Deleted:
Evans, 200 | (Langhamer et al., 2018; Mayr et al., 2018; Smith &
17) |
| Deleted:               | optimizing (Newton-Raphson algorithm)                      |
| Deleted:               |                                                            |
| Deleted:               | determined                                                 |
| Deleted:               | measurements                                               |
| Deleted:               | 0.00052 m m -1 (                                |
| Deleted:               | 2                                                          |
| Deleted:               | )                                                          |
| Deleted:               | smaller                                                    |

[revised manuscript text omitted]

---

## Referee Report (RR1)

**Review by Marius Schaefer, 6[th] of January 2020**

**General comments:**

The revised version of the manuscript shows several improvements in comparison with the original submission. However many of the points which I mentioned in my first assessment remained unadressed. To my point of view the model description section I still weak and the presentation of the results and their validation is insufficient. Both points could be easily improved (see detailed comments below). The general readability of the manuscript could be improved by separating the presentation of the results from the discussion.

**Detailed comments:**

**Metodology:**

**DR-scaling:** it is still not clear to me how the vertical precipitation gradient is "optimized". Which function are you minimizing or looking for zeros using the Newton-Raphson algorithm (  better know as Newton method)? I would very much recommend to perform this method with two drying ratios ( 0.45 and 0.6) to be able to compare systematically with the OPM.

**Linear orographic precipitation model:**

Some equation are shown now, but in my point of view they rather confuse than contribute. Several variables are not explained. If you want to show equations, I would very much prefer to see the two coupled advection equations for cloud water and hydrometeors. They contain the models essentials. Then you can describe the solution methods with words ( or equations if you prefer) and give the necessary references. You should comment on how the choice of every model parameter influences the results. For the second experiment I would recommend to only change the drying ratio to 0.6 and leave the other model parameters untouched in order to get a better idea about the influence of this parameter ( then you could also avoid the word extreme scenario).

**Results**

**Physical constrains on local precipitation.**

In this section I would like to see all modeled precipitation fields (Figure 3) to the see the differences in precipitation distribution of the DR-scaling and OPM (similar as it was done in Weidemann 2013). Precipitation totals seem to be very similar between both methods (Table 1).
And I would like to see an analysis ( hopefully one or more figures) indicating which of the set-ups
 is reproducing best the station data and which are possible biases (a differentiated analysis of stations on the windward and leeward side of the icefields can indicate a lot about the models performance to simulate the process of orographic precipitation.)

**Consequences of revised precipitation estimates on the surface mass balance of the SPI**

Again, since this is a regional assessment, you should include the consequences of your modeled precipitation fields for both icefields.There is a very clear relationship between accumulation and surface mass balance for NPI as well. See figure below and attached data.

[Figure]

At NPI calving losses are much smaller and SMB can be much better validated against geodetic surveys ( such as Braun, 2019).

---

## Author Response (AR2)

I thank the reviewer for the constructive comments and suggestions. My response to the review can be found in the attached document.

**R**: Referee's comment
**A**: Author's response

**General**

I have revised the manuscript according to the reviewers' comments. There is a new subsection in Section 2 describing the experiments. The descriptions of the DR-scaling and OPM have been updated and partly re-written. As requested by the reviewer, the impact of the revised precipitation on the SMB of the NPI is now discussed in Section 4.2 and further considered in all following sections. For better overview, the paragraph about the constrains of the hydrological cycle on the SMB has been moved to a new Section (4.3). In the course of the revision, two new figures (Fig. 4 and 5) have been added. Also, Table 1 shows the precipitation estimates from the DR-scaling experiments, now.
* * *
**DR-scaling**

**R**: DR-scaling: it is still not clear to me how the vertical precipitation gradient is "optimized". Which function are you minimizing or looking for zeros using the Newton-Raphson algorithm (better known as Newton method)? I would very much recommend to perform this method with two drying ratios (0.45 and 0.6) to be able to compare systematically with the OPM.

**A**: I have expanded the paragraph about the DR-scaling and its optimization in the hope that the procedure is now more comprehensible. For better illustration, I have included the optimization equation. In addition, precipitation was estimated using both $DRS_{0.45}$ and $DRS_{0.60}$ (please note that I have introduced the abbrevations $OPM_{0.45}$, $OPM_{0.60}$, $DRS_{0.45}$, and $DRS_{0.60}$ for the experiments in the new Section 2 (Experiments), now). The sentences anticipating the results from these experiments have been moved to the first paragraph of chapter 3.2. In this paragraph, among other things, the optimized lapse-rate and the precipitation estimate for the $DRS_{0.60}$ estimation is now presented. Furthermore, Table 1 has been extended with the corresponding data.
* * *
**Linear orographic precipitation model**

**R**: Some equation are shown now, but in my point of view they rather confuse than contribute. Several variables are not explained. If you want to show equations, I would very much prefer to see the two coupled advection equations for cloud water and hydrometeors. They contain the models essentials. Then you can describe the solution methods with words ( or equations if you prefer) and give the necessary references. You should comment on how the choice of every model parameter influences the results. For the second experiment I would recommend to only change the drying ratio to 0.6 and leave the other model parameters untouched in order to get a better idea about the influence of this parameter (then you could also avoid the word extreme scenario).

**A**: I have revised the description of the OPM (Section 2) and kept only the fundamental equations. The theoretical foundations and implementation of the model as well as in-depth sensitivity studies have been described in detail in a vast number of studies, e.g. Smith and Barstad (2004), Bartstad and Smith (2005), Jiang and Smith (2003), Jiang (2003), Smith and Evans (2007), Kunz and Kottmeier (2006) etc. I therefore refrain from repeating the fundamental equations here and refer to the changes made for this study. However, I agree with the reviewer that not all quantities have been explained in the necessary detail and have done so in this manuscript version.

As far as the fixing of the Brunt-Väisälä frequency and the sensitivity factor is concerned, I do not agree with the reviewer. This strategy would not be in line with the aim of this study. The aim was to create an 'extreme' (ensemble) scenario that reflects observed atmospheric conditions but does not occur in its frequency. To achieve this, these quantities were deliberately set to the 98th percentile to derive the highest possible precipitation estimate. As already mentioned above, the influence of the individual quantities has already been investigated in theoretical sensitivity studies.

References:

Barstad I and Smith RB (2005) *Evaluation of an orographic precipitation model*. J. Hydromet., 6(1), 85–99.
Jiang Q (2003) *Moist dynamics and orographic precipitation*. Tellus A, 55(4), 301–316.
Jiang Q and Smith RB (2003) *Cloud timescales and orographic precipitation*. J. Atmos. Sci., 60(13), 1543–1559.
 Kunz M and Kottmeier C (2006) *Orographic enhancement of precipitation over low mountain ranges. Part I: Model formulation and idealized simulations*. J. Appl. Meteorol. Climatol., 45(8), 1025–1040.
Smith RB and Barstad I (2004) *A linear theory of orographic precipitation*. J. Atmos. Sci., 61(12), 1377–1391.
Smith RB and Evans JP (2007) *Orographic precipitation and water vapor fractionation over the Southern Andes*. J. Hydromet., 8(1), 3–19.
* * *
**Physical constraints on local precipitation**

**R**: In this section I would like to see all modeled precipitation fields (Figure 3) to the see the differences in precipitation distribution of the DR-scaling and OPM (similar as it was done in Weidemann 2013). Precipitation totals seem to be very similar between both methods (Table 1). And I would like to see an analysis ( hopefully one or more figures) indicating which of the set-ups is reproducing best the station data and which are possible biases (a differentiated analysis of stations on the windward and leeward side of the icefields can indicate a lot about the models performance to simulate the process of orographic precipitation.)

**A**: I have added another figure (Fig. 4) now that shows the differences between OPM and DR-scaling (although I think this figure is not necessary since the DR-scaling reflects, per definition, the topography). The one to one comparison between simulations and observations is given in detail in Table S3 in the supplement. For sake of clarity, I have added a new Figure 5 visualizing the relation between observations and the outcome of the different experiments. The figure (and the provided statistics such as bias and rmse) clearly confirms that $OPM_{0.45}$ outperforms the $OPM_{0.60}$ simulation. Again, there are only three station located west of the Patagonian Icefields which complicates an in-depth comparison. These stations are highlighted in Figure 5 to emphasize the windward and leeward performance.
* * *
**Consequences of revised precipitation estimates on the surface mass balance of the SPI**

**R**: Again, since this is a regional assessment, you should include the consequences of your modeled precipitation fields for both icefields. There is a very clear relationship between accumulation and surface mass balance for NPI as well. See figure below and attached data.

**A**: Thank you very much for the dataset. I have extended the analysis and discussion to the NPI throughout the text accordingly.

[revised manuscript text omitted]

**Contents of this file**

[Figure]

**Figure S1: Schematic illustration of the atmospheric large-scale circulation and moisture transport in the South Pacific. Shown are the location of the westerlies for austral summer (December-January-February), the barrier jet along the Andes (red arrows), and the mean moisture transport by baroclinic eddies (blue shaded arrow). The shading indicates regions of high water vapor variability.**

[Figure]

**Figure S2: Monthly IWV anomalies in Puerto Montt ( a) and Punta Arenas ( b). Shown are the running 3-month mean IWV anomalies for the atmospheric soundings and the nearest ERA-Interim grid point from 1990-2016. The blue shaded areas indicate very strong El Niño events (ONI>1.5).**

[Figure]

**Figure S3: Mean differences in the IWV between ERA Interim and SSMI data for the period 1988-2016. Red shading indicates a positive bias in the ERA Interim data.**

[Figure]

**Figure S4: Linear trend of IWV in the SSM/I and ERA-Interim data for the period 1988-2016 (in % per decade). Dotted areas indicate significant long-term trends (p<0.05).**

[Figure]

5  **Figure S5: Frequency distribution of the non-dimensional mountain height. The non-dimensional mountain height is calculated from ERA-Interim data (2010-2016) off Patagonia's west coast (Fig. 1, D1).**

**Table S1.** Automatic weather stations in Patagonia. Precipitation sums and trends are given in mm. Bold numbers indicate significant trends (p<0.05).

| Station | Lat | Lon | Sum | Trend in % per decade | from | to |
|---|---|---|---|---|---|---|
| Pirihueico En Pirihueico | -40.02 | -71.72 | 2875 | **-0.36** | 1999 | 2015 |
| El Llolly | -40.07 | -72.62 | 1763 | -0.02 | 1995 | 2015 |
| Catamutun | -40.17 | -73.17 | 1858 | 0.02 | 1998 | 2015 |
| Venecia | -40.19 | -73.43 | 3964 | -0.04 | 1998 | 2015 |
| Lago Maihue | -40.22 | -72.15 | 2982 | **-0.17** | 1980 | 2015 |
| Trinidad | -40.31 | -73.43 | 1713 | 0.13 | 1998 | 2015 |
| Lago Ranco | -40.32 | -72.47 | 1923 | 0.01 | 1980 | 2015 |
| Adolfo Matthei | -40.59 | -73.11 | 1256 | -0.05 | 1983 | 2016 |
| Canal Bajo Osorno Ad. | -40.61 | -73.06 | 1232 | -0.04 | 1980 | 2016 |
| Anticura | -40.66 | -72.18 | 1185 | -0.39 | 1998 | 2015 |
| Rio Negro En Chahuilco | -40.71 | -73.23 | 1240 | -0.21 | 2004 | 2016 |
| Futacuhuin | -40.72 | -72.44 | 1681 | -0.07 | 1995 | 2015 |
| Rupanco | -40.77 | -72.68 | 1665 | -0.04 | 1994 | 2015 |
| San Antonio Oeste Aero | -40.78 | -65.10 | 202 | **0.25** | 1980 | 2016 |
| Gobernador Castello | -40.87 | -63.00 | 272 | **0.21** | 1980 | 2016 |
| Purranque | -40.94 | -73.14 | 1309 | 0.00 | 1999 | 2015 |
| Frutillar | -41.13 | -73.06 | 1467 | 0.00 | 1994 | 2015 |
| San Carlos De Bariloche | -41.15 | -71.16 | 589 | **0.21** | 1980 | 2016 |
| Fresia | -41.15 | -73.41 | 1613 | -0.03 | 1994 | 2015 |
| La Ensenada | -41.23 | -72.57 | 2358 | 0.00 | 1980 | 2005 |
| Maquinchao | -41.25 | -68.73 | 96 | 0.25 | 1980 | 2016 |
| Lago Chapo | -41.42 | -72.60 | 3038 | -0.05 | 1999 | 2014 |
| El Tepual Puerto Montt Ap. | -41.44 | -73.10 | 1590 | -0.05 | 1980 | 2016 |
| Puerto Montt | -41.46 | -72.94 | 1845 | 0.00 | 1980 | 2016 |
| Puerto Montt | -41.46 | -72.94 | 1845 | 0.00 | 1980 | 2016 |
| Maullin | -41.62 | -73.60 | 1668 | -0.04 | 1987 | 2015 |
| Puelo | -41.65 | -72.31 | 2642 | -0.17 | 1997 | 2016 |
| Ancud 1 (Dga) | -41.86 | -73.82 | 2013 | 0.12 | 1993 | 2015 |
| Hornopiren | -41.94 | -72.44 | 2971 | **-0.49** | 1998 | 2016 |
| El Bolson | -41.94 | -71.53 | 232 | **1.02** | 1996 | 2016 |
| Chepu | -42.05 | -73.97 | 2554 | -0.08 | 1999 | 2015 |
| Quemchi | -42.14 | -73.47 | 2437 | -0.03 | 2000 | 2015 |
| Castro 1 (Dga) | -42.46 | -73.77 | 1555 | **-0.37** | 1993 | 2008 |
| Cucao | -42.62 | -74.11 | 2067 | 0.05 | 1997 | 2015 |
| Chaiten | -42.91 | -72.71 | 3408 | 0.26 | 1998 | 2007 |
| Chaiten Ad. | -42.93 | -72.70 | 3590 | -0.02 | 1980 | 2007 |
| Esquel Aero | -42.93 | -71.15 | 372 | **0.18** | 1980 | 2016 |
| Quellon | -43.11 | -73.61 | 1781 | 0.01 | 1993 | 2015 |
| Puerto. Cardenas | -43.18 | -72.43 | 3922 | -0.33 | 2001 | 2015 |
| Futaleufu Ad. | -43.19 | -71.85 | 1958 | -0.01 | 1980 | 2016 |

| | | | | | |
|---|---|---|---|---|---|
| Trelew Aero | -43.20 | -65.27 | 171 | 0.11 | 1980 | 2016 |
| Lago Espolon | -43.22 | -71.93 | 2693 | -0.13 | 2001 | 2015 |
| Valle Rio Frio | -43.47 | -72.35 | 3712 | **-0.24** | 2001 | 2015 |
| Christchurch Intl | -43.49 | 172.53 | 486 | **0.05** | 1980 | 2016 |
| Cape Bruny Lighthouse | -43.49 | 147.15 | 940 | -0.07 | 1980 | 2016 |
| Alto Palenaad. | -43.61 | -71.81 | 1589 | -0.04 | 1980 | 2016 |
| Palena | -43.62 | -71.78 | 1597 | **-0.20** | 2001 | 2015 |
| Maatsuyker Island Lighthouse | -43.66 | 146.27 | 1235 | **-0.16** | 1980 | 2016 |
| Marin Balmaceda | -43.77 | -72.95 | 2388 | 0.17 | 1994 | 2015 |
| Paso De Indios | -43.82 | -68.88 | 86 | 0.44 | 1980 | 1996 |
| Chatham Islands Aws | -43.95 | -176.57 | 624 | 0.16 | 1994 | 2011 |
| La Junta | -43.97 | -72.41 | 1998 | -0.07 | 1981 | 2016 |
| Bordalit | -44.05 | -72.32 | 2508 | **-0.29** | 1994 | 2011 |
| Lago Verde | -44.24 | -71.85 | 1244 | 0.41 | 1994 | 2011 |
| Puerto Puyuhuapi | -44.32 | -72.56 | 2994 | -0.08 | 1981 | 2015 |
| Rio Cisnes | -44.50 | -71.31 | 273 | **-0.26** | 1981 | 2015 |
| La Tapera | -44.65 | -71.67 | 687 | -0.29 | 1981 | 2002 |
| Cisnes Medio | -44.67 | -72.27 | 2162 | 0.00 | 1982 | 2015 |
| Puerto Cisnes | -44.73 | -72.68 | 2812 | -0.09 | 1981 | 2015 |
| Villa Maihuales | -45.17 | -72.15 | 1399 | -0.09 | 1986 | 2015 |
| Estancia Bao Nuevo | -45.27 | -71.53 | 490 | -0.01 | 2001 | 2015 |
| Irehuao | -45.27 | -71.71 | 409 | 0.11 | 1994 | 2015 |
| Villa Ortega | -45.37 | -71.98 | 654 | -0.14 | 1981 | 2015 |
| Puerto Aysen Ad. | -45.40 | -72.66 | 2011 | -0.11 | 1980 | 2016 |
| Puerto Aysen | -45.40 | -72.70 | 2197 | **-0.32** | 1994 | 2008 |
| El Balseo | -45.40 | -72.49 | 1745 | -0.06 | 1981 | 2015 |
| Rio Aysen En Puerto Aysen | -45.41 | -72.62 | 1944 | 0.00 | 2003 | 2016 |
| Puerto Chacabuco | -45.46 | -72.82 | 2678 | -0.05 | 1985 | 2015 |
| Coyhaique Alto | -45.48 | -71.60 | 221 | 0.02 | 1985 | 2016 |
| Coyhaique Conaf | -45.55 | -72.06 | 887 | -0.19 | 2003 | 2015 |
| Rio Simpson Bajo Junta Coyhaique | -45.55 | -72.07 | 723 | -0.24 | 2006 | 2016 |
| Coyhaique (Escuela Agricola) | -45.57 | -72.03 | 808 | -0.06 | 1984 | 2016 |
| Teniente Vidal Coyhaique Ad. | -45.59 | -72.11 | 986 | 0.03 | 1980 | 2016 |
| Comodoro Rivadavia | -45.78 | -67.50 | 203 | 0.00 | 1980 | 2016 |
| Balmaceda Ad. | -45.91 | -71.69 | 529 | -0.03 | 1980 | 2016 |
| Villa Cerro Castillo | -46.12 | -72.15 | 540 | **-0.41** | 1993 | 2015 |
| Rio Ibaez En Desembocadura | -46.27 | -71.99 | 490 | 0.26 | 2006 | 2016 |
| Puerto Ibaez | -46.29 | -71.93 | 396 | **0.47** | 1986 | 2008 |
| Invercargill Airpor | -46.42 | 168.33 | 888 | 0.01 | 1980 | 2016 |
| Alfred Faure (Iles Crozet) | -46.43 | 51.85 | 1652 | 0.02 | 1980 | 2016 |
| Bahia Murta | -46.46 | -72.67 | 1208 | **-0.21** | 1994 | 2015 |
| Perito Moreno Arpt | -46.52 | -71.02 | 74 | -0.08 | 1980 | 2016 |
| Chile Chico | -46.54 | -71.71 | 228 | -0.13 | 1994 | 2015 |
| Chile Chico Ad. | -46.58 | -71.69 | 255 | -0.01 | 1980 | 2016 |
| Puerto Guadal | -46.84 | -72.70 | 565 | 0.31 | 1994 | 2015 |
| Marion Island | -46.88 | 37.87 | 1857 | **-0.12** | 1980 | 2016 |

| | | | | | |
|---|---|---|---|---|---|
| Estancia Valle Chacabuco | -47.12 | -72.48 | 189 | -0.29 | 1994 | 2015 |
| Rio Baker En Angostura Chacabuco | -47.14 | -72.73 | 745 | 0.05 | 2004 | 2016 |
| Lord Cochrane Ad. | -47.24 | -72.59 | 702 | -0.07 | 1980 | 2016 |
| Rio Cochrane En Cochrane | -47.25 | -72.56 | 480 | 0.19 | 2005 | 2016 |
| Puerto Deseado | -47.74 | -65.90 | 66 | -0.25 | 1980 | 2012 |
| Caleta Tortel | -47.80 | -73.54 | 1987 | -0.13 | 2003 | 2016 |
| Rio Pascua Ante Junta Rio Quetru | -48.16 | -73.09 | 2082 | -0.07 | 2004 | 2016 |
| Rio Mayer Reten | -48.21 | -72.32 | 200 | 1.07 | 1994 | 2003 |
| Villa Ohiggins | -48.47 | -72.56 | 771 | **0.36** | 1994 | 2008 |
| Lago Ohiggins En Villa Ohiggins | -48.52 | -72.60 | 895 | -0.08 | 2004 | 2016 |
| Gobernador Gregores | -48.78 | -70.17 | 63 | -0.85 | 1980 | 1996 |
| Candelario Mancilla | -48.88 | -72.74 | 458 | 0.05 | 1994 | 2016 |
| Puerto Eden | -49.12 | -74.41 | 2346 | **-0.43** | 1998 | 2010 |
| San Julian | -49.31 | -67.80 | 190 | 0.09 | 1980 | 2016 |
| Port-Aux-Francais (Iles Kergu | -49.35 | 70.25 | 591 | 0.02 | 1980 | 2016 |
| El Calafate Aero | -50.27 | -72.05 | 145 | -0.16 | 2004 | 2016 |
| Lago Argentino Arpt | -50.33 | -72.30 | 120 | **0.98** | 1980 | 1999 |
| Enderby Island Aws | -50.48 | 166.30 | 737 | 0.07 | 1993 | 2016 |
| Lago Dickson | -50.82 | -73.11 | 910 | **0.93** | 2004 | 2016 |
| Cerro Guido | -50.90 | -72.33 | 265 | **0.10** | 1984 | 2016 |
| Rio Las Chinas En Cerro Guido | -51.05 | -72.52 | 226 | 0.29 | 2005 | 2016 |
| Torres Del Paine | -51.18 | -72.97 | 739 | 0.04 | 1983 | 2016 |
| Cerro Castillo | -51.26 | -72.33 | 315 | 0.00 | 1981 | 2016 |
| Rio Gallegos Aero | -51.62 | -69.28 | 214 | **0.19** | 1980 | 2016 |
| Teniente Gallardo Puerto Natales Ad. | -51.67 | -72.53 | 192 | **1.69** | 1999 | 2016 |
| Casas Viejas | -51.70 | -72.33 | 256 | **0.24** | 1981 | 2015 |
| Puerto Natales | -51.73 | -72.48 | 490 | -0.02 | 1986 | 2016 |
| Mount Pleasant | -51.82 | -58.45 | 471 | 0.03 | 1992 | 2016 |
| Rio Rubens En Ruta N 9 | -52.03 | -71.94 | 414 | **-1.29** | 2007 | 2016 |
| Teniente Merino | -52.03 | -70.73 | 221 | 0.09 | 1984 | 2015 |
| Rubens En Ruta N. 9 | -52.04 | -71.94 | 473 | **-0.52** | 1990 | 2005 |
| Rio Penitente En Morro Chico | -52.05 | -71.42 | 265 | -0.15 | 2007 | 2016 |
| Monte Aymond | -52.16 | -69.61 | 250 | -0.03 | 1996 | 2016 |
| Villa Tehuelche | -52.44 | -71.40 | 337 | -0.04 | 1981 | 2016 |
| Rio Perez | -52.55 | -71.96 | 525 | -0.04 | 1990 | 2014 |
| Campbell Island Aws | -52.55 | 169.17 | 1017 | **0.44** | 1996 | 2016 |
| Seno Skyring | -52.55 | -71.96 | 584 | -0.56 | 2002 | 2016 |
| San Gregorio | -52.57 | -70.07 | 264 | -0.03 | 1992 | 2016 |
| Rio Verde | -52.60 | -71.50 | 337 | 0.02 | 1994 | 2016 |
| Rocallosas | -52.65 | -71.96 | 327 | **-0.88** | 1994 | 2015 |
| Cerro Sombrero | -52.78 | -69.29 | 248 | -0.06 | 1984 | 2014 |
| Bahamondes | -52.80 | -72.93 | 3367 | **-0.75** | 2000 | 2015 |
| Bahia San Felipe | -52.87 | -69.93 | 351 | 0.12 | 1980 | 2016 |
| Isla Riesco | -52.88 | -71.57 | 408 | -0.09 | 1991 | 2015 |
| Punta Arenas | -53.12 | -70.88 | 923 | 0.02 | 1980 | 2016 |
| Canal De Trasvase Estero Llau-Llau | -53.13 | -70.94 | 648 | -0.10 | 2005 | 2016 |

| | | | | | | |
|---|---|---|---|---|---|---|
| Rio Las Minas En Bt. Sendos | -53.14 | -70.99 | 780 | -0.03 | 2000 | 2016 |
| Las Minas | -53.14 | -70.98 | 774 | -0.14 | 1996 | 2015 |
| Laguna Lynch | -53.14 | -70.98 | 435 | -0.04 | 1980 | 2015 |
| Fuentes Martinez Porvenir Ad. | -53.19 | -70.32 | 260 | -0.03 | 1986 | 2016 |
| Porvenir | -53.29 | -70.37 | 315 | **-0.20** | 1991 | 2015 |
| Onaisin En Maria Cristina | -53.31 | -69.27 | 315 | -0.12 | 1990 | 2016 |
| San Sebastian | -53.32 | -68.66 | 294 | 0.13 | 1990 | 2016 |
| Lago Parrillar | -53.40 | -71.25 | 800 | 0.00 | 1990 | 2016 |
| Cameron | -53.64 | -69.65 | 366 | 0.08 | 1994 | 2016 |
| San Juan | -53.65 | -70.96 | 544 | **0.29** | 1980 | 2016 |
| Russfin | -53.76 | -69.19 | 429 | -0.01 | 1994 | 2016 |
| Rio Caleta En Tierra Del Fuego | -53.86 | -70.00 | 340 | -0.15 | 2007 | 2016 |
| Rio Grande En Tierra Del Fuego | -53.89 | -68.88 | 276 | 0.14 | 2007 | 2016 |
| Seccion Rio Grande | -53.90 | -68.92 | 401 | 0.15 | 1991 | 2011 |
| Pampa Huanaco | -54.05 | -68.80 | 340 | **0.28** | 1994 | 2016 |
| Macquarie Island | -54.50 | 158.94 | 912 | -0.03 | 1980 | 2016 |
| Ushuaia Malvinas Argentinas | -54.84 | -68.30 | 335 | 0.24 | 1980 | 2016 |
| Guardia Marina Zanartu Pto Williams | -54.93 | -67.62 | 466 | -0.09 | 1980 | 2016 |
| Rio Robalo En Puerto Williams | -54.95 | -67.64 | 389 | 0.14 | 2005 | 2016 |

**Table S2. Comparison of the IWV trends between atmospheric soundings and the nearest ERA-Interim grid point. Numbers are given in mm decade$^{-1}$. Bold numbers indicate significant trends ($p<0.05$).**

| | 1988-2016 | 1988-2009 | 2010-2016 |
|---|---|---|---|
| **Puerto Montt** | | | |
| Radiosounding | -0.22 (-1.5%) | -0.42 (-3.0%) | 2.87 (19.4%) |
| ERA-Interim | -0.16 (-1.3%) | -0.25 (-2.0%) | 0.65 (5.2%) |
| | | | |
| **Punta Arenas** | | | |
| Radiosounding | 0.23 (2.2%) | 0.07 (0.6%) | 1.23 (11.3%) |
| ERA-Interim | 0.14 (1.3%) | 0.05 (0.4%) | 0.80 (7.0%) |

**Table S3. Comparison of the OPM$_{0.45}$ and OPM$_{0.60}$ experiments (DR=0.45) with observations. Given are the latitude (lat), longitude (lon), altitude (alt), the precipitation values from the orographic precipitation model experiments (OPM$_{0.45}$ and OPM$_{0.60}$) and the observations (Obs) at the weather stations. The precipitation values are given in mm yr$^{-1}$.**

| Location | lat | lon | alt | OPM$_{0.45}$ (0.45) | OPM$_{0.60}$ (0.6) | Obs |
|---|---|---|---|---|---|---|
| Villa Cerro Castillo | -46.12 | -72.15 | 345 | 471 | 693 | 282 |
| Rio Ibaez En Desembocadura | -46.26 | -71.99 | 220 | 298 | 463 | 623 |
| Bahia Murta | -46.46 | -72.66 | 240 | 849 | 1275 | 1017 |
| Lago General Carrera Fachinal | -46.54 | -72.22 | 18 | 579 | 823 | 333 |
| Glaciar San Rafael | -46.64 | -73.85 | 8 | 3829 | 5198 | 1271 |
| Puerto Guadal | -46.84 | -72.70 | 210 | 745 | 1141 | 656 |

| | | | | | | |
|---|---|---|---|---|---|---|
| Estancia Valle Chacabuco | -47.11 | -72.48 | 343 | 613 | 874 | 159 |
| Rio Nef Antes Junta Estero El Revalse | -47.13 | -73.08 | 281 | 895 | 1405 | 974 |
| Rio Baker En Angostura Chacabuco | -47.14 | -72.72 | 160 | 673 | 1029 | 856 |
| Lago Cachet 2 En Glaciar Colonia | -47.19 | -73.25 | 427 | 1088 | 1676 | 243 |
| Lord Cochrane Ad. | -47.24 | -72.58 | 204 | 744 | 1056 | 652 |
| Rio Cochrane En Cochrane | -47.25 | -72.56 | 140 | 788 | 1094 | 514 |
| Rio Colonia En Nacimiento | -47.33 | -73.11 | 146 | 986 | 1526 | 1261 |
| Caleta Tortel | -47.79 | -73.53 | 10 | 2389 | 3318 | 1870 |
| Rio Pascua Ante Junta Rio Quetru | -48.15 | -73.08 | 20 | 1668 | 2342 | 2137 |
| Lago Ohiggins En Villa Ohiggins | -48.51 | -72.59 | 300 | 752 | 1084 | 909 |
| Candelario Mancilla | -48.87 | -72.73 | 300 | 850 | 1276 | 519 |
| Rio Punta Eva En Puerto Eden | -49.11 | -74.41 | 10 | 3607 | 5429 | 2840 |
| El Calafate Aero | -50.26 | -72.05 | 204 | 433 | 586 | 149 |
| Lago Dickson | -50.82 | -73.11 | 200 | 1301 | 1906 | 1130 |
| Lago Paine | -50.84 | -72.90 | 440 | 1132 | 1671 | 500 |
| Cerro Guido | -50.89 | -72.33 | 230 | 820 | 1097 | 312 |
| Amalia | -50.95 | -73.69 | 0 | 4350 | 5539 | 2801 |
| Rio Paine En Parque Nacional 2 | -50.96 | -72.79 | 90 | 1139 | 1675 | 724 |
| Nunatak Grey | -50.97 | -73.22 | 300 | 1414 | 2059 | 589 |
| Lago Sarmiento | -51.01 | -72.71 | 110 | 1111 | 1633 | 352 |
| Lago Pehoe | -51.07 | -72.99 | 40 | 1347 | 1966 | 868 |
| Lago Grey | -51.11 | -73.13 | 50 | 1397 | 2064 | 663 |
| Glaciar Tindall | -51.11 | -73.28 | 345 | 1584 | 2282 | 1200 |
| Torres Del Paine | -51.18 | -72.96 | 25 | 1431 | 2106 | 750 |
| Rio Rincon En Ruta Y-290 | -51.31 | -72.82 | 36 | 1554 | 2285 | 769 |
| Rio Serrano En Desembocadura | -51.33 | -73.10 | 25 | 1671 | 2456 | 1227 |

**Table S4. Summary of the WRF configuration.**

| | Value |
|---|---|
| **Domain configuration** | |
| Horizontal grid spacing | 12.5-km, 2.5-km, and 500-m |
| Vertical levels | 55 |
| Model top pressure | 100 hPa |
| | |
| **Model physics** | |
| Radiation | RRTMG |
| Microphysics | Morrison |
| Cumulus | Kain-Fritsch |
| Planetary boundary layer | MYNN Level 2.5 |
| Atmospheric surface layer | Monin Obukhov |

| | |
|---|---|
| Land surface | Noah-MP |
| Top boundary condition | Rayleigh damping |

**Lateral boundaries**

| | |
|---|---|
| Forcing | ERA-Interim |
| | 0.75°x0.75°, 6-hourly |